# Capture and Reuse of Carbon Dioxide (CO₂) for a Plastics Circular Economy: A Review

**Laura Pires da Mata Costa** [1] , **Débora Micheline Vaz de Miranda** [1] , **Ana Carolina Couto de Oliveira** [2] , **Luiz Falcon** [3] , **Marina Stella Silva Pimenta** [3] , **Ivan Guilherme Bessa** [3] , **Sílvio Juarez Wouters** [3] , **Márcio Henrique S. Andrade** [3] and **José Carlos Pinto** [1,*]

1   Programa de Engenharia Química/COPPE, Universidade Federal do Rio de Janeiro, Cidade Universitária, CP 68502, Rio de Janeiro 21941-972, Brazil; laura@peq.coppe.ufrj.br (L.P.d.M.C.); dmiranda@peq.coppe.ufrj.br (D.M.V.d.M.)

2   Escola de Química, Universidade Federal do Rio de Janeiro, Cidade Universitária, CP 68525, Rio de Janeiro 21941-598, Brazil; acco@eq.ufrj.br

3   Braskem S.A., Rua Marumbi, 1400, Campos Elíseos, Duque de Caxias 25221-000, Brazil; falcon@braskem.com (L.F.); marina.pimenta@braskem.com (M.S.S.P.); ivan.bessa@braskem.com (I.G.B.); silvio.wouters@braskem.com (S.J.W.); marcio.andrade@braskem.com (M.H.S.A.)

*   Correspondence: pinto@peq.coppe.ufrj.br; Tel.: +55-21-3938-8709

**Abstract:** Plastic production has been increasing at enormous rates. Particularly, the socioenvironmental problems resulting from the linear economy model have been widely discussed, especially regarding plastic pieces intended for single use and disposed improperly in the environment. Nonetheless, greenhouse gas emissions caused by inappropriate disposal or recycling and by the many production stages have not been discussed thoroughly. Regarding the manufacturing processes, carbon dioxide is produced mainly through heating of process streams and intrinsic chemical transformations, explaining why first-generation petrochemical industries are among the top five most greenhouse gas (GHG)-polluting businesses. Consequently, the plastics market must pursue full integration with the circular economy approach, promoting the simultaneous recycling of plastic wastes and sequestration and reuse of CO₂ through carbon capture and utilization (CCU) strategies, which can be employed for the manufacture of olefins (among other process streams) and reduction of fossil-fuel demands and environmental impacts. Considering the previous remarks, the present manuscript's purpose is to provide a review regarding CO₂ emissions, capture, and utilization in the plastics industry. A detailed bibliometric review of both the scientific and the patent literature available is presented, including the description of key players and critical discussions and suggestions about the main technologies. As shown throughout the text, the number of documents has grown steadily, illustrating the increasing importance of CCU strategies in the field of plastics manufacture.

**Keywords:** carbon dioxide (CO₂); plastic; carbon capture and utilization (CCU); circular economy; plastics manufacture

## 1. Introduction

Plastics are here to stay. These versatile materials are used in broad ranges of applications in very distinct fields, including packaging for goods and food, fibers and films, clothes, pharmaceutical and biomedical materials, healthcare products, components for vehicles, and pieces for the construction industry, among many others [1]. For this reason, plastic consumption has been increasing since World War II (see Figure 1). In 2019, over 368 Mt of polymers were produced worldwide, including thermoplastics, thermosets, polyurethanes, elastomers, adhesives, coatings and sealants, and polypropylene (PP) fibers [2]. Most commercial plastics are fossil-based [3], which, according to the World Economic Forum (WEF), accounts for 4–8% of the annual global oil consumption, used for heat generation (50%) and for the production process (50%) [4]. As the annual plastic

production is expected to grow to 1323 Mt in 2050, the plastics business is expected to account for at least 20% of the annual global oil consumption in 2050 [5–7]. Even more impressive, the International Energy Agency (IEA) estimates that half of the annual global oil production will be destined for the manufacture of plastics in 2060 [8]. In addition, projections show that, even with the decrease of the worldwide consumption of plastics in 2020 due to Covid-19, the petrochemical business will continue to grow due to demand-side dependencies on global supply chains [9–11]. Moreover, it is expected that, in spite of the bans imposed on single-use plastics, the related decrease in oil consumption will be very modest [12]. The pervasive consumption of plastics comes with a price, which is now being detected in the form of plastic waste generation and emission of polluting gases.

Before discussion of the technical issues, it is important to present some assumptions and terminologies that are adopted in the present manuscript. First, it is necessary to acknowledge that several studies revealed the existence of a direct relationship between climate change and carbon dioxide ($CO_2$) emissions [13]. Second, $CO_2$ emissions can be evaluated with help of the carbon footprint (CF), which is as a tool that is used to calculate the total amount of greenhouse gas (GHG) released by an organization, event, or product [14] and can be interpreted as an indicator of sustainable development [15]. Third, it is also assumed that the global warming impact (GWI) of organizations, events, or products can be expressed in terms of carbon dioxide equivalent (or $CO_2e$), which represents a common unit for description of different GHG [16]. For example, carbon dioxide ($CO_2$), methane ($CH_4$), and nitrous oxide ($N_2O$) present relative GWIs of 1, 25, and 298 $CO_2$-equivalents, respectively [17,18].

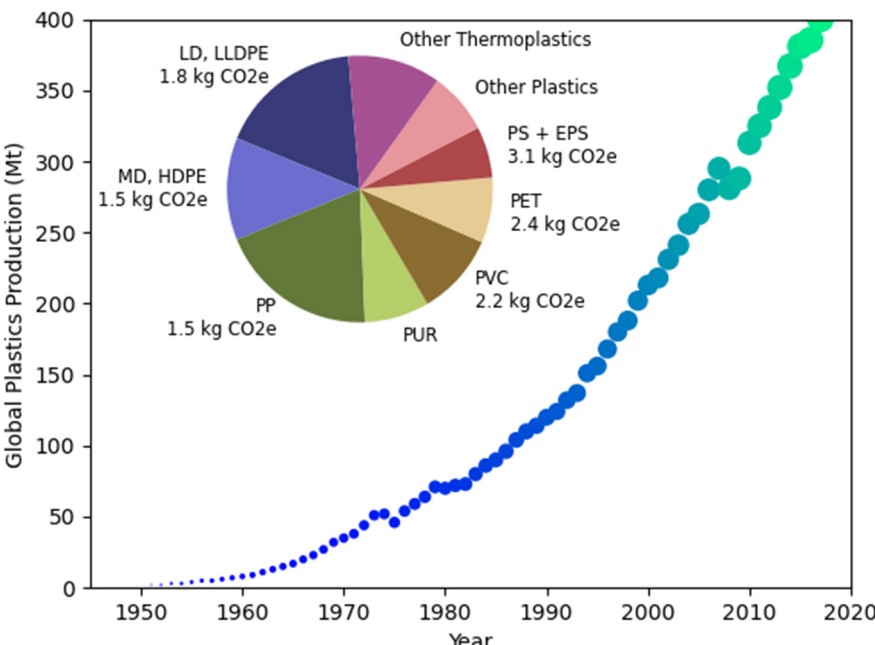

**Figure 1.** World plastic production throughout the years (1950–2019) [5,19–23] and the most produced resins in 2019, with the respective global warming impacts (GWIs) [7]. Mt: megaton; PS: polystyrene; EPS: expandable polystyrene; PP: polypropylene; MD, HDPE: medium-density/high-density polyethylene; LLDPE: linear low-density polyethylene; LDPE: low-density polyethylene; PUR: polyurethane; PET: poly(ethylene terephthalate); PVC: poly(vinyl chloride); other plastics include acrylonitrile butadiene styrene resin (ABS), polybutylene terephthalate (PBT), polycarbonate (PC), polymethyl methacrylate (PMMA), and polytetrafluoroethylene (PTFE).

Figure 1 presents the relative global production of distinct plastics in 2019 with the respective GWIs [7]. According to the Center for International Environmental Law (CIEL), the total lifecycle GHG emissions of plastics are expected to grow from 0.86 $GtCO_2e$ in 2019 to 2.80 $GtCO_2e$ in 2050 [6]. Moreover, cumulative emissions between 2015 and 2050 may

exceed 56 Gt in this field, equivalent to 14% of the carbon budget of the whole industrial sector [6,24]. Alternatively, Zheng and Suh (2019) estimated an even worse scenario, where the plastic GHG emissions would increase from 1.8 $GtCO_2e$ in 2015 to 6.5 $GtCO_2e$ in 2050, when partial incineration, recycling, and landfill disposal were considered as alternatives for final disposal of plastic wastes (resulting in 4.1 metric tons of $CO_2e$ per metric ton of plastic resin produced from fossil fuels) [7]. It must also be considered that the IEA estimated a more disadvantageous scenario, affirming that plastic production emitted 2.2 $GtCO_2e$ in 2016 [12].

Despite the available predictions, there is no unequivocal evaluation of the magnitude of the overall lifecycle GHG emissions of plastics. It is usually assumed that the plastic industries are among the four most GHG-polluting industries (iron, steel, cement, and plastics), which are responsible for 66% of the total industrial $CO_2e$ emissions [8,25–28]. However, due to the steeper increase of plastic consumption, the plastic sector is also expected to present the largest growth of $CO_2$ emissions [24,29–31] in the forthcoming years, which constitutes a matter of concern [15,32]. It must be highlighted that plastic carbon footprints necessarily include the extraction or manufacture of the raw materials, the transformation process, the product distribution, the particular type of product consumption, and the final product disposal, as all these stages release carbon into the atmosphere.

As a whole, the fast growth of consumption, short lifecycles, and incorrect final disposal of many plastic products have already caused significant environmental impacts that must be mitigated and corrected. Around 79% of the total plastic waste ever produced is accumulating in landfills, dumps, or the natural environment, while 12% has been incinerated and only 9% has been recycled [5,32]. Consequently, 91% of the total plastic material produced throughout the last decades are still degrading slowly in the environment and in landfills, which does not make any practical sense.

Particularly, these degradation scenarios do not provide any significant value to the production chain, especially because polymer materials are able to capture, store, and recycle carbon [17]. As a matter of fact, it is important to note that conventional fossil-based plastics are considered one of the most sustainable materials, despite all the problems described previously [17,33]. Many studies have indicated that the use of plastic materials constitutes an efficient choice in terms of energy consumption and GWI because of several inherent key characteristics, which make them highly competitive with other common materials (such as glass, metal, wood, and paper), encouraging the continued growth of consumption of plastic products [1]. Considering these conflicting aspects, it becomes possible to suggest that the main problem in the plastics industry is the proper integration of the whole plastics chain with the approaches of a circular economy, helping to combat the ever-intensifying climate change, encouraging the use of renewable feedstocks and renewable energy, promoting the recycling of wastes [17,25], and, as discussed in the present review, using carbon capture and utilization strategies to circulate the carbon atoms in the plastics chain [2,34–37].

## 1.1. Carbon Footprint and the Plastics Industries

Naphtha and natural gas liquids (NGLs) are the main sources of fossil-based feedstocks for the petrochemical and chemical industries, including ethylene, propylene, butadiene, butylene, and aromatics [8,38–40]. While the use of NGLs is more common in the United States of America (USA) and, more recently, in other countries that are importing chemicals from the USA [41], naphtha is the main source of feedstocks in the Middle East, Europe, and South America.

Steam cracking (SC) is the technology used most often to transform naphtha or NGL into smaller hydrocarbon molecules, such as ethylene and propylene [6,18,42]. However, SC also is the most energy-intensive process in the chemical and petrochemical industries [27,38], consuming roughly 65% of the total energy demand of both ethane- and naphtha-based olefin plants [42,43]. For instance, Johansson et al. (2012) reported that most $CO_2$ emissions in a Borealis cracker could be associated with the cracker furnaces,

while additional $CO_2$ emissions could be associated with the boilers used to produce high- and medium-pressure steam for the cracker plant, the hot oil furnace, and flaring [44]. Catalytic cracking (CC) is an emerging alternative technology employed to increase the olefins yield, but is also highly energy-demanding, representing 20% to 50% of the total $CO_2$ emissions of typical refineries due to the necessity to regenerate the catalyst used in the process [45]. As a whole, summing the $CO_2$ emissions associated with fuel combustion and utilities, it has been calculated that olefin production emits approximately 0.4–1.2 kg $CO_2$e and 1.6–1.8 kg $CO_2$e per kg of olefin produced by ethane and naphtha cracking processes, respectively [30,42,46–48]. Particularly, ethane cracking requires lower temperatures and leads to higher hydrogen and ethylene contents, emitting less $CO_2$ per metric ton of produced ethylene than naphtha cracking processes [6,42,49]. Despite that, many previous studies neglected the amounts of $CO_2$ emissions associated with monomer manufacture for evaluation of life-cycle analyses (LCA) of plastic products, considering the monomer as the starting point and overlooking the emissions caused by extraction, refining, and cracking processes, making the quantification of GHG emissions of the plastic chain doubtful [50].

Due to the increasing availability of stranded gas, catalytic dehydrogenation (DH or DDH) of light alkanes has also become a well-established process for manufacture of light olefins. This is particularly true for manufacture of propylene through catalytic dehydrogenation (PDH) due to the relatively low yields of propene in NGL cracking [38,51]. Other olefin production technologies are also being studied and developed, including coal-to-olefins [52], oxidative coupling of methane (OCM) [38], syngas-based routes (including the Fischer–Tropsch synthesis, FTS, and the methanol synthesis followed by methanol-to-olefin, MTO), using both coal or NGL as feedstocks [53], or the direct conversion of crude oil-to-olefin (CTO) [54]. Unfortunately, these processes are still much more expensive than cracking processes and are not yet well established [38]. Additional information about these technologies is presented in Section 4 of the present manuscript.

GHG emissions can also be associated with the production stages of the plastic resin chain (the second generation plants of the fossil-based plastic chain, as illustrated in Figure 2), as the polymerization and processing (including extrusion, injection molding, and blow molding) steps also demand significant amounts of heat and power, although these production stages demand lesser amounts of energy than the first-generation plants. For example, while the pyrolysis furnaces of most steam crackers operate at temperatures ranging from 700 to 950 °C, typical ethylene polymerization reactors operate below 150 °C, such that 37–43% of the $CO_2$e emissions of final PE (HDPE, LDPE or LLDPE) pellets can be associated with the polymerization stage [55,56].

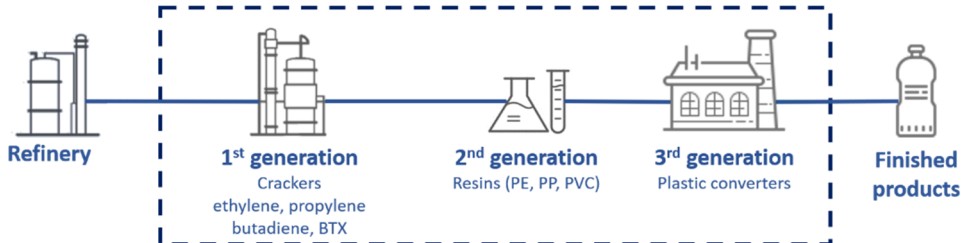

**Figure 2.** Schematic representation of a typical fossil-based plastic chain.

It must be emphasized that the collection of information regarding $CO_2$ emissions can be very complex, as emissions depend on the sources of the employed raw materials, the inputs needed for manufacture of utilities and the particular technologies applied at each plastic plant. For instance, the CIEL reported that *"direct greenhouse gas emissions from petrochemical and resin manufacturers typically depend on facility efficiency, configuration, and age, the desired end product or product mix, preferred feedstocks, fuel sources, and regulatory constraints*

*and compliance (such as emissions limits, requirements for emissions control technologies or practices, and enforcement)"* [6].

Most of the GHG emissions in the plastic chain are related to energy (electricity and heat) demand, such that about 90% of $CO_2$ emissions can be related to energy consumption [43]. Electricity normally comes from the electrical grid and is usually not produced in the site [57]. Heat, on the contrary, comes from burning of fossil fuels locally [58]. However, direct GHG process emissions can also be important and should be carefully accounted for, as in the cases of cracking processes and monomer production steps. For illustrative purposes, Figure 3 shows simplified process schemes for the manufacture of polyethylene and polypropylene from both fossil (using naphtha as feedstock) and bio-based ethanol, presenting the respective typical energy and heat requirements. A detailed version of Figure 3 is shown in Figure A1. Figure 3 also presents the direct $CO_2$ emissions, which are complemented by data presented in Table 1 [56]. As one can observe, the amounts of generated $CO_2$ and the complexity to capture $CO_2$ from process streams change, due to variations of the $CO_2$ concentrations, presence of other gases, and technological level of the plant. For instance, according to Table 1, the ethylene oxide production stage does not emit much $CO_2$, because $CO_2$ produced in side reactions in this case can be captured in a parallel recovery plant [59]. It must be emphasized that additional steps not described in Figure 3 and Table 1 can be required to produce some polymers. For example, polystyrene manufacture requires the production of styrene from benzene and ethylene, while manufacture of PET requires the manufacture of ethylene glycol from ethylene oxide (which comes from ethylene and oxygen) and terephthalic acid from xylene and acetic acid [6], consuming petrochemicals and catalysts while emitting more $CO_2$.

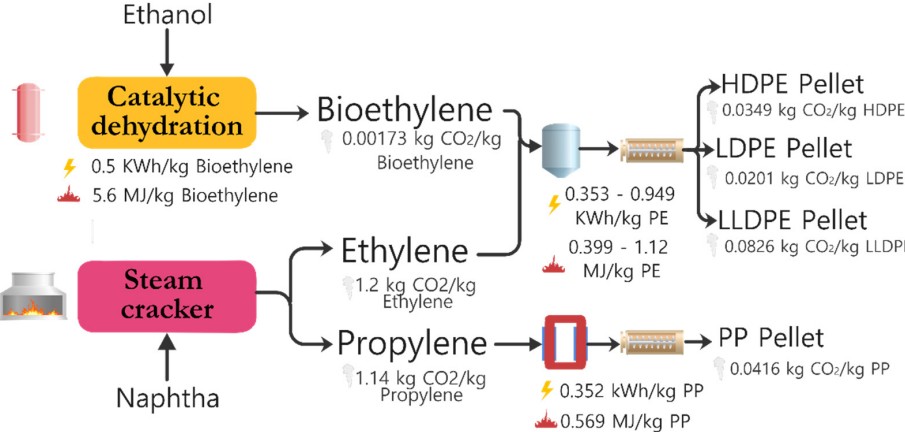

**Figure 3.** Energy consumption and $CO_2$ emissions for polyethylene and polypropylene production chains (data from the Ecoivent 3.6 dataset [56]). A detailed version is available in Appendix A.

**Table 1.** Carbon dioxide emissions in the production of some typical feedstocks (data from the Ecoivent 3.6 dataset [56]).

|  | Ethylene | Ethylene Oxide | Ethylene Glycol | Propylene | Xylene | Terephthalic Acid |
|---|---|---|---|---|---|---|
| $kg\,CO_2/kg\,product$ | 1.12 | 0.21 | 0 | 1.14 | 1.3 | 0.106 |
|  | Benzene | Vinyl chloride | Hydrogen | Styrene | HDPE | LDPE |
| $kg\,CO_2/kg\,product$ | 1.12 | 0.187 | 0.114 | 0.114 | 0.0349 | 0.0020 |
|  | LLDPE | PP | PET | PS | HIPS | PVC |
| $kg\,CO_2/kg\,product$ | 0.0826 | 0.0416 | 0 | ■ | ■ | 0.0108 |

■ not available in the Ecoinvent 3.6 dataset.

As the high energy demand is responsible for the biggest share of $CO_2$ emissions [48,49,60,61] in the plastic chain, improving energy efficiency and changing the energetic sources from fossil-based to renewable alternatives (wind power, solar power, nuclear energy, and renewable natural gas derived from landfill gas or biomass) can exert a significant impact on the environmental performances of plastic processes, leading to more sustainable technological routes and a long-term reduction in GHG emissions [49,61–69]. For instance, assuming the consumption of 100% renewable energy (wind power and renewable natural gas), the GHG emissions of plastic production in the USA could be reduced by 50–75% [58]. Moreover, according to Zheng and Suh (2019), assuming the annual growth rate of 4% for plastic production, the use of 100% renewable energy sources can reduce the GHG emissions of the plastics chain by 62% until 2050, even if fossil-fuel sources remain the only feedstock employed for future production of plastics [7]. The total emissions can be reduced even further (77%) if all post-consumed plastics are recycled, making use of 100%-renewable energy sources [7]. As one can see, the impact of shifting to renewable energy can be so big that this finding encouraged the creation of the *"Cracker of the Future"* consortium, including BASF, Borealis, BP, LyondellBasell, SABIC, and Total. The consortium aims at the complete electrification of steam crackers, replacing the use of fossil fuels by renewable electricity [70].

Nonetheless, switching to renewable energy sources is still far from happening because of the low prices of fossil fuels, lack of governmental incentives, lack of robust and efficient batteries, and necessity to adapt most industrial facilities [7,36]. For these reasons, Amghizar et al. (2020) proposed alternatives to reduce up to 30% of the $CO_2$ emissions within the current energy scenario, including the development of novel furnace designs, enhancement of heat transfer capabilities, and reduction in coke formation [43]. Moreover, given other process-related $CO_2$ emissions, strategies are also needed to reduce the environmental impact of manufacture processes. For example, considering the 100% renewable energy scenario, resin production based on fossil fuels would be responsible for 86% of the GHG emissions, resulting in 0.9–1.1 $kgCO_2e$ per kg plastic [3,4]. Therefore, development of more selective catalysts, implementation of heat integration schemes, and reduction in energy leaking to the environment, among other energy and materials saving technologies, should be pursued by plastic manufacturers.

A third strategy that must be seriously considered is changing the feedstocks to biomass. In particular, a significant advantage of biobased polymers is that these products can be entitled to GHG credits for the amount of biogenic carbon stored in the polymer [71], which can render the total GHG emissions negative during a cradle-to-gate analysis [7,58,71,72]. In Brazil, where the biobased plastic production is already established, the impact of GHG emissions can be even lower due to the higher availability of renewable energy sources and excellent conditions for growing of sugarcane and other sources of biomass [7,72,73]. In spite of that, a residual impact of GHG emissions would remain (roughly 0.5 and 0.2 $kgCO_2e$ per kg of plastic for corn and sugarcane, respectively), given the $CO_2$ emissions associated with corn or sugarcane cultivation and harvesting, ethanol fermentation and distillation, and chemical transformations (dehydration of ethanol and oxidization of ethylene to ethylene oxide) [7,71,74]. In fact, if all post-consumed plastics are recycled, assuming the use of 100% renewable energy sources, GHG emissions can be reduced by 84% and 86% when corn- and sugarcane-based plastics are considered [7,58]. Nevertheless, it is relevant to notice that the use of renewable energy is far more impactful than changing the feedstock to renewable alternatives [7,58]. Other biobased technologies can be employed to produce ethylene, making use of enzymes and microorganisms, but these technologies are not mature enough and not yet economically viable [75,76].

Implementing a circular economy approach can also constitute an important strategy to reduce the carbon footprint of the plastic chain, since circular strategies can keep the carbon circulating in the production chain for long periods, extracting the maximum value of the raw materials and recovering and regenerating products and materials at the end of their service life [4,77], as illustrated in Figure 4. Chemical recycling processes, such

as pyrolysis (thermal or catalytic depolymerization of plastic wastes at 300–600 °C in absence of oxygen), among others, can be employed to convert post-consumed plastics into fresh feedstocks, reducing the necessity to produce fossil fuels and the overall GHG emissions. For example, according to estimates, complete recycling of plastic wastes could save 3.5 billion barrels of oil per year [78]. Moreover, 1 metric ton of LDPE produced from pyrolysis oil emits 2.3 metric tons of $CO_2$e less than the same material produced from fossil naphtha [79–83]. It is important to emphasize that, when plastic wastes are not recycled, the slow degradation leads to $CO_2$ emissions that are usually not accounted for in most analyzed scenarios [84]. Furthermore, it is important to consider the use of 100% renewable energy sources and to integrate the pyrolysis unit to the petrochemical chain in order to reduce the GHG impact of chemical recycling processes [85].

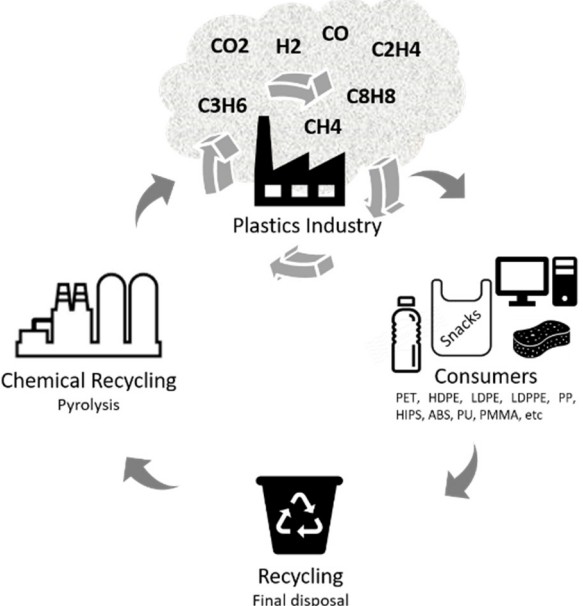

**Figure 4.** Schematic representation of a carbon circular economy approach for the plastic chain.

*1.2. The Role of Carbon Capture*

Since the increase in plastic production is evident, as documented in several recent works and reports [1,5–7,86], and considering the environmental problems associated with the standard linear economy model, where plastics (mostly intended for single-use) are disposed fast through improper methods [17,32,87], strategies must be implemented to reduce the plastic carbon footprint. While the use of renewable feedstocks and energy sources might constitute part of the solution [88], additional actions should be taken to reduce the industrial GHG emissions and to develop efficient recycling technologies [89–91]. In addition to the increase in sustainability of the plastic chain, these activities should consider the following [92]:

i.      response to society demands;
ii.     development of innovative processes and products;
iii.    attainment of circular economy conditions through chemical recycling of plastic wastes and carbon;
iv.     production of liquid fuels and integration within the existing refining infrastructure;
v.      attainment of the United Nations Sustainability Development Goals (SDGs), in order to ensure access to affordable, reliable, sustainable, and modern energy; build resilient infrastructure; promote inclusive and sustainable industrialization and foster innovation; make cities and human settlements inclusive, safe, resilient, and sustainable; take urgent action to combat climate change and its impacts;
vi.     improvement of the public image of plastic products;

vii.    obtainment of emission reduction credits.

A possible strategy to fully include the plastic chain into the circular economy scheme involves the sequestration and reutilization of emitted carbon dioxide through carbon capture and utilization (CCU) techniques [82–86]. When most of the carbon circulates in the plastic chain, the demands for fossil fuels and the environmental impacts of the operations are diminished. Although some CCU techniques may not yet be economically profitable, the shift toward circular economy approaches and sustainable processes and products is evidenced by developing technologies, creation of carbon taxes, and marketing [28,93–98]. Meanwhile, the global screening of industrial players and a greater involvement of discussions about the use of CCU technologies in the plastic chain are still scarce in the literature. For these reasons, the main purpose of this manuscript is to connect carbon dioxide emissions with the use of CCU techniques in the plastics industry through a detailed bibliometric review of the scientific and patent literature.

## 2. Methodology

An investigation of the scientific and patent literature regarding the general capture and use of carbon dioxide can be indeed very extensive. Given the vast number of available documents and the scope of interest, a bibliometric search strategy was proposed, as presented in Figure 5. The proposed bibliometric approach comprised four phases: (i) initial search of the many general reviews available in the literature (Phase 1a); (ii) selection of papers related to the plastic chain (Phase 1b); (iii) identification of active industrial players and active startups (Phase 2a); (iv) analysis of the current technological status of the field, according to a survey of available patents (Phase 2b).

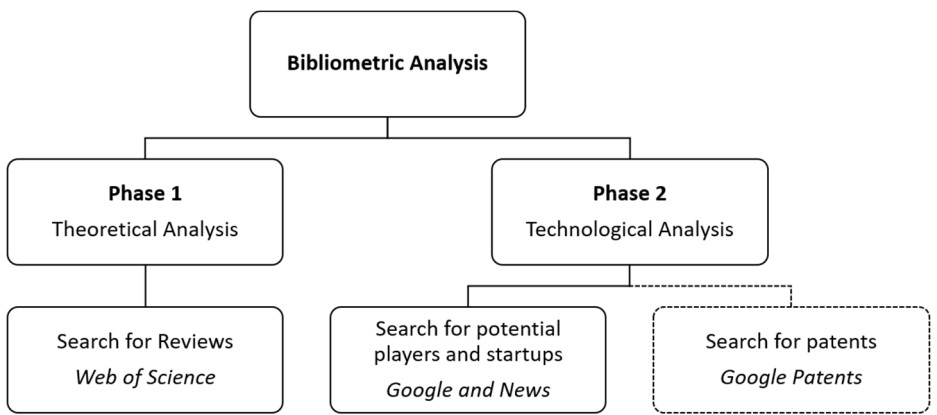

**Figure 5.** Flowchart of the proposed bibliometric search approach.

### 2.1. Search Methods and Procedures (Phase 1)

*Web of Science* was selected as the initial search platform, as it provides access to a wide range of world-class scientific literature associated with the investigated field, ensuring the high quality of the proposed search [99]. The initial search results were obtained between 18 September and 23 October 2018. After that, the database was continuously enlarged with additional references obtained through the previously collected manuscripts and published in scientific journals, as monitored through periodical search updates and browsing.

The keyword sets employed in the initial search and posterior search updates were ((carbon OR "carbon dioxide") AND (captur* OR sequestr*) AND (recycl* OR feedstock OR recover* OR re-use OR reuse) AND (usage OR use OR using OR utili?ation OR convers* OR transfor* OR process* OR manage*) AND (fuel* OR chemical* OR asset* OR product* OR solvent* OR fertilizer* OR plastic* OR foam* OR paint* OR coating* OR mineral* OR gas*)). Keywords were searched in the title, abstract, and keywords sections of the manuscripts. As the initial search provided 8810 documents, the initial analysis was concentrated on the many reviews available in the field. Considering the scope of the present work, papers related to $CO_2$ storage, $CO_2$ capture from soils and forests, genetic manipulation of

microorganisms and genetic characterization of biomass, water treatment, $CO_2$ mitigation (legislation), LCA studies of direct $CO_2$ utilization, and mineralization and carbonation of $CO_2$ were discarded. Following this procedure, 325 reviews were selected for initial theoretical analysis and screening of technologies. The reviews were then classified as "capture", "utilization", or "capture and utilization", as shown in Figure 6. Lastly, the papers related more specifically to the plastic chain were included in the database. The selected documents were downloaded and the topics described in Table 2 were analyzed and recorded.

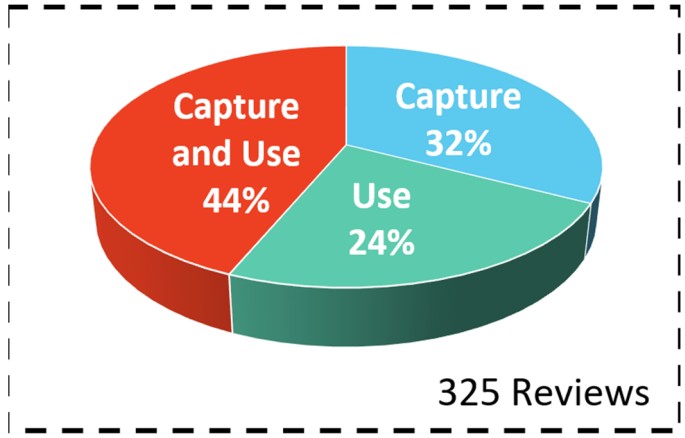

**Figure 6.** Distribution of selected reviews.

**Table 2.** Topics selected for bibliometric analyses of scientific papers.

| General Information | Source | Capture | Use |
|---|---|---|---|
| Title | Source | Capture technique | Conversion technique |
| Authors | Quantity | Method | Method |
| Institution | Concentration | Material | Conversion product |
| Year | | Experimental conditions | Efficiency |
| Country | | Supplier | Experimental conditions |
| Journal | | Concentration of | Application |
| Number of citations | | Cost | Material |
| Main subject: capture/use | | Volume of | Catalyst |
| | | Selectivity | Reactor |
| | | Recovery | Reaction mechanisms |
| | | Operating scale | Operating scale |
| | | Relevant references | Supplier |
| | | Maturity | Maturity |

## 2.2. Search Methods and Procedures (Phase 2)

The technological analysis comprised two phases. First, the screening of potential players was performed through extensive searches on the internet, especially in the news sections of search engines. Keywords were selected as described in the previous section. Then, identified players, including startups, were selected, and their patents were searched for in the *Google Patents* database. Afterward, the search for patents was completed with additional searches in the *Google Patents* database using the same keywords described before. As described in the previous section, the downloaded patents were also classified as "*capture*", "*use*", and "*capture and use*". The identified players and startups are listed in Table 3. The initial search results were obtained between 15 October 2018 and 6 March 2019. After that, the database was continuously enlarged with additional patents obtained through the previously collected patents and monitoring through periodical search updates. The selected documents were downloaded and the topics described in Table 4 were analyzed and recorded.

**Table 3.** Identified players and startups that are active in the field of carbon capture and utilization (CCU) techniques for the plastics industry.

| Players (A–C) | Players (C–L) | Players (L–Z) |
|---|---|---|
| Algatec | C-Capture | **LG Chem** |
| A2BE Carbon Capture | CERT | LiquidLight |
| AGG Biofuel | **Chevron Phillips** | **Lotte Chemical** |
| Air Co. | Climeworks | **LyondellBasel** |
| Algae AquaCulture Technology | $CO_2$ Solutions | **Mitsubishi Chemical** |
| Algenol Biofuels | $CO_2$ Concrete | MOF Technologies |
| Aljadix | Cool Planet Energy Systems | Mosaic Materials |
| Alpha | Covestro | Net Power |
| BASF | Dimensional Energy | Newlight Technologies |
| Blue Planet | Dioxide Materials | Novomer |
| Bodega Algae | **Dow Chemical** | OPUS12 |
| Borouge | EcoEra | **PetroChina** |
| Breathe | Econic Technologies | Phycal |
| Bright Energy | **ENI** | $PHYCO_2$ |
| bse engineering | EnobraQ | Phytonix |
| C12 Energy | Evolution Petroleum | Pond Technologies |
| C2CNT | **ExxonMobil** | **Reliance Industries** |
| C4X | **Formosa Plastics Corp** | $RenewCO_2$ |
| CalciTech | FuelCell Energy | **Sabic** |
| Calera | Global Thermostat | Sinopec |
| CarbFix carbon mineralization | Green Minerals | Skyonic |
| Carbicrete | Grow Energy | SkyTree |
| Carbon Capture Machine | Hago energetics | **Sumitomo Chemical** |
| Carbon Clean Solutions | **INEOS** | Sustainable Energy Solutions |
| Carbon Engineering | InnoSepra | Synthetic Genomics |
| Carbon Recycling International | Innovator Energy | Tandem Technical |
| Carbon Upcycling Technologies Inc. | Jupiter Oxygen | **Toray Industries** |
| CarbonCure | Just BioFiber Structural Solutions Corp. | Trelys |
| Carbonfree Chemicals (Skyonic Corporation) | **Lanxess** | |
| Carbozyme | LanzaTech | |

**In bold**: Top primary plastics (first-generation petrochemicals) producers. Data source: [100,101].

**Table 4.** Topics selected for bibliometric analyses of patents.

| Player Specification | Search Strategy | Patent Information |
|---|---|---|
| Player/assignee | Keywords search | Code |
| Official site | Number of results | Applicant/assignee |
| Activities | | Author(s) |
| Sector | | Priority year |
| Country | | Status |
| Logo | | Title |
| | | Main subject (capture and/or use) |

## 3. Results

### 3.1. Bibliometric Analysis of Scientific Review Papers: General Information

3.1.1. The Annual Distribution

As shown in Figure 7, the number of publications in this field has been growing continuously. For example, up to 2018, the number of review papers regarding capture or use of carbon dioxide as feedstock increased approximately 10% per year. Then, the number decreased in 2019, before growing again in 2020. It is important to emphasize

that this large number of published papers, reviews in particular, is highly uncommon in other areas.

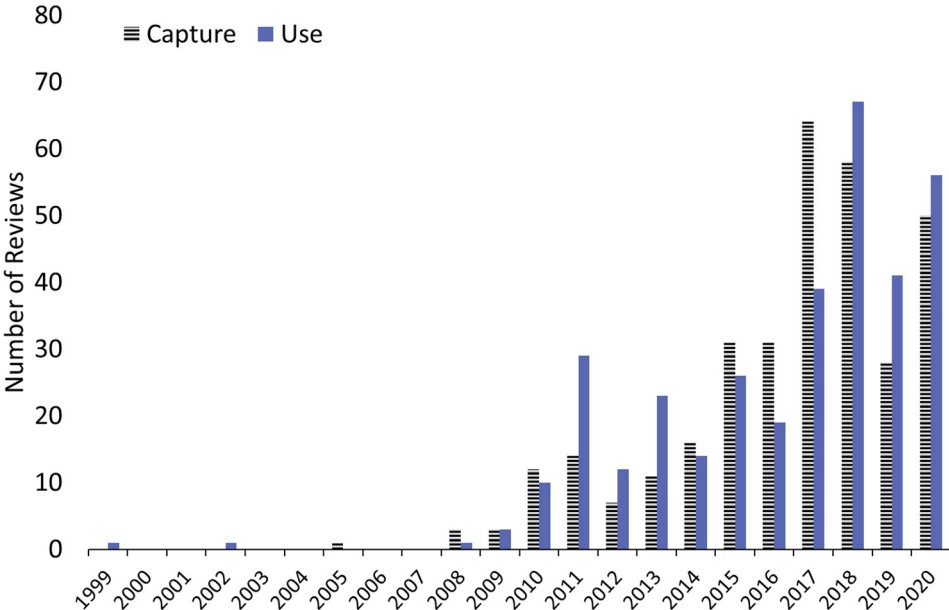

**Figure 7.** Distribution of published review papers regarding carbon capture and use, related to the main scope of the present work.

The first review published in this area dates back to 1991 [102], certainly motivated by the installation of the Intergovernmental Panel on Climate Change, established by the United Nations Environment Program and the World Meteorological Organization in 1988, which eventually led to the Kyoto Protocol, in 1992, leveraging researches in carbon dioxide ($CO_2$), methane ($CH_4$), nitrous oxide ($N_2O$), hydrofluorocarbons (HFCs), perfluorocarbons (PFCs), and sulfur hexafluoride ($SF_6$) emissions. However, the interest in $CO_2$ capture did not start with the greenhouse effects concern, but was promoted by the potential economic benefits of using captured $CO_2$ for enhanced oil recovery (EOR) operations, when $CO_2$ is injected into oil reservoirs to increase the mobility of the oil and, therefore, the productivity of the well [103].

Herzog and Drake (1996) [104] wrote the first review of capture and use of carbon dioxide technologies, employing processes based on chemical absorption using monoethanolamine (MEA) as an absorber, the main method used by commercial plants in the 1990s. However, research was already mature in areas related to integrated gasification combined cycle (IGCC). In this review, the use of $CO_2$ for production of plastics was mentioned as a possibility.

Figure 8 presents the distribution of the researched patents during the last 38 years. The growing number of patents available in this field can be easily noticed. However, it can also be observed that the main subject of interest has shifted slowly from capture to use of $CO_2$, which constitutes perhaps the main challenge for industrial sites and chemical companies. Apparently, companies are still searching for good opportunities to use the carbon dioxide streams made available by capture technologies. However, many $CO_2$ processes involving polyols, methanol, and electrolysis have become important in recent years.

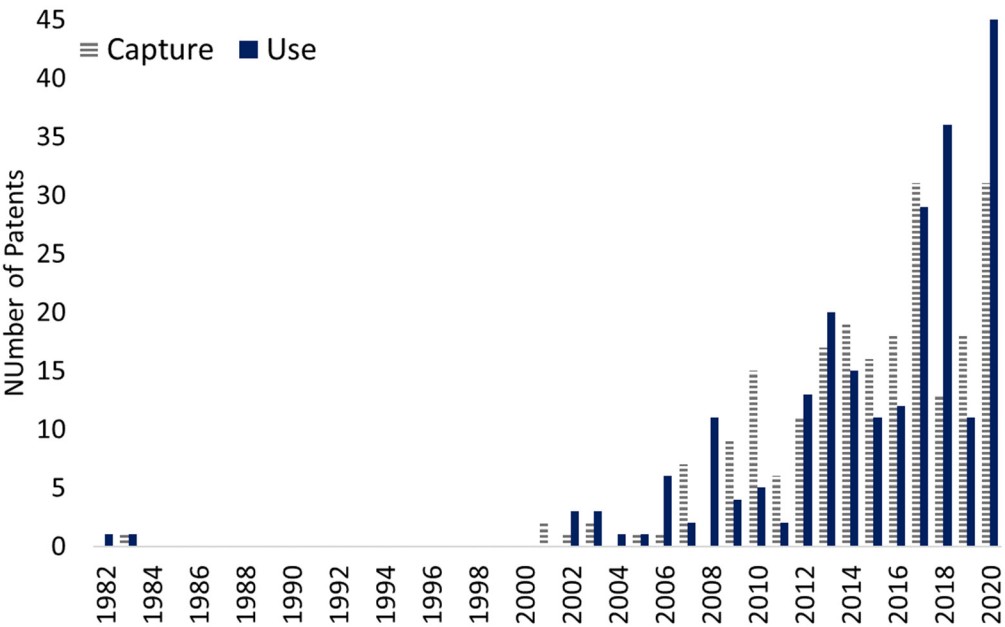

**Figure 8.** Distribution of published patents regarding carbon capture and use.

### 3.1.2. The Scientific Journals Distribution

The distribution of publications in scientific journals is shown in Table 5, for journals that published four or more papers in the analyzed area. As one can see, publications have been concentrated in relatively few journals, with a significant concentration in *"Renewable and Sustainable Energy Reviews"* (which published 27 reviews or 15% of the analyzed dataset). The high quality of the journals (with impact factor (IF) values above 3.0) must be highlighted and indicates that this issue is regarded as very relevant by the academic community.

### 3.1.3. The Country and Institutions Distribution

Figure 9 illustrates the distribution of review publications around the world. It can be observed that the USA (19%) and China (12%) are leading countries in publications about carbon dioxide capture and/or use. The most productive institutions in the field are also located in these countries, as observed in Table 6; however, it is not possible to highlight a few particular institutions as the most productive ones this subject area. The institutions with the largest number of published review papers related to the scope of the present work were the Nankai University and University of Malaya, with four reviews each. However, this represents only 2.2% of all surveys available in the literature. It can be observed that the publications of the most productive institutions represent only approximately 17% of the analyzed review papers, reinforcing again that it is a diverse and diffuse field of study.

### 3.1.4. The Most Cited Review Papers

Table 7 shows the top five most cited papers in the investigated field. As one can observe, the most cited papers describe the carbon capture and/or use employing different technologies. This probably shows that the scientific research in this area is still driven by the necessity to develop new technologies, although this can also be attributed to the specific solutions and problems that can be associated with each particular approach and technology. It is also possible to notice that the top five most cited papers were published after 2000, which means that this can still be regarded as an emerging subject, as evidenced in Tables 7 and 8.

**Table 5.** Distribution of review papers in scientific journals in the field of CCU.

| Ranking | Journal | IF [1] | NP [2] | Percentage (%) |
|---|---|---|---|---|
| 1 | Renewable and Sustainable Energy Reviews | 10.556 | 27 | 15.0 |
| 2 | Progress in Energy and Combustion Science | 26.467 | 6 | 3.3 |
| 3 | Bioresource Technology | 6.669 | 5 | 2.8 |
| 4 | Energy and Environmental Science | 33.250 | 5 | 2.8 |
| 5 | International Journal of Greenhouse Gas Control | 3.231 | 5 | 2.8 |
| 6 | Journal of Utilization | 5.189 | 5 | 2.8 |
| 7 | Energy and Fuels | 3.021 | 4 | 2.2 |
| 8 | Journal of Cleaner Production | 6.395 | 4 | 2.2 |

[1] IF: impact factor (2018); [2] NP: number of publications.

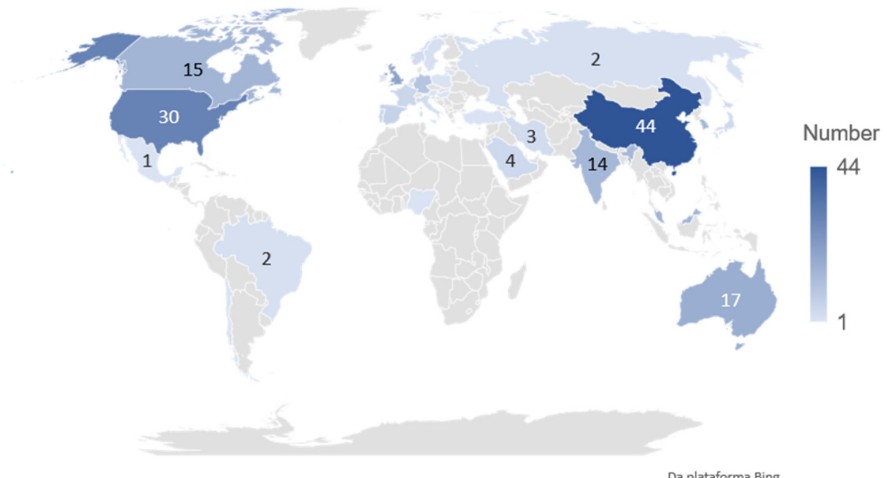

**Figure 9.** Country distribution of reviews in the field of CCU.

**Table 6.** Most productive institutions regarding the scope of this work.

| Ranking | Institution | Country | Review Publications | Percentage (%) | Subject |
|---|---|---|---|---|---|
| 1 | Nankai University | China | 4 | 2.2 | Capture and use |
| 2 | University of Malaya | Malaysia | 4 | 2.2 | Capture and use |
| 3 | Universiti Teknologi Malaysia | Malaysia | 3 | 1.7 | Capture and use |
| 4 | South China University of Technology | China | 3 | 1.7 | Capture and use |
| 5 | Tsinghua University | China | 3 | 1.7 | Capture and use |
| 6 | Monash University | Australia | 3 | 1.7 | Capture and use |

**Table 6.** *Cont.*

| Ranking | Institution | Country | Review Publica-tions | Percentage (%) | Subject |
|---|---|---|---|---|---|
| 7 | Hunan University | China | 3 | 1.7 | Capture and use |
| 8 | Hanyang University | Republic of Korea | 3 | 1.7 | Capture and use |
| 9 | Chinese Academy of Sciences | China | 3 | 1.7 | Capture and use |

**Table 7.** Top five most cited review papers associated with the scope of the present work.

| # | Paper | Subject | Technology | Citations | Ref |
|---|---|---|---|---|---|
| 1 | An overview of current status of carbon dioxide capture and storage technologies Authors: Leung, D. Y., Caramanna, G., and Maroto-Valer, M. M. Source: Renewable and Sustainable Energy Reviews (2014) | Capture and use | Absorption Adsorption CLC [1] Membranes Gas hydrates Cryogenic distillation Chemical conversion EOR [3] and/or ECBM [2] | 1697 | [105] |
| 2 | An overview of $CO_2$ capture technologies Authors: MacDowell, N., Florin, N., Buchard, A., et al. Source: Energy & Environmental Science (2010) | Capture | Membranes Adsorption CLC [1] | 1300 | [106] |
| 3 | Sequestration of carbon dioxide in coal with enhanced coalbed methane recovery: A review Authors: White, C. M., Smith, D. H., Jones, K. L., et al. Source: Energy and Fuels (2005) | Capture | Adsorption ECBM [2] | 918 | [107] |
| 4 | Anthroporgenic Chemical Carbon Cycle for a Sustainable Future Authors: Olah, G. A., Prakash, G. S., and Goeppert, A. Source: Journal of the American Chemical Society (2011) | Capture and use | Absorption Pre-combustion Hydrogenation Coal gasification Electrochemical processes | 907 | [108] |
| 5 | Sustainable hydrocarbon fuels by recycling $CO_2$ and $H_2O$ with renewable or nuclear energy Authors: Graves, C., Ebbesen, S. D., Mogensen, M., and Lackner, K. S Source: Renewable and Sustainable Energy Reviews (2011) | Use | Electrochemical processes | 846 | [109] |

[1] CLC: chemical looping combustion; [2] ECMB: enhanced coalbed methane; [3] EOR: enhanced oil recovery.

**Table 8.** Main sources of $CO_2$. Data reproduced with permission from Karim Ghaib, Fatima-Zahrae Ben-Fares, Renewable and Sustainable Energy Reviews; published by Elsevier, 2018 [110]; and Hasmukh A Patel, Jeehye Byun, Cafer T Yavuz, ChemSusChem, published by John Wiley and Sons, 2017 [111].

| Sector | $CO_2$ Sources | $CO_2$ Concentration in Exhaust Gas (%vol) |
|---|---|---|
| | Biomass fermentation | 15–50 |
| Biomass processes | Biogas upgrading | ~100 |
| | Bioethanol production | ~100 |
| | Natural gas combustion | 3–5 |
| Power generation plants | Petroleum combustion | 3–8 |
| | Coal combustion | 10–15 |
| | Cement production | 14–33 |
| | Iron and steel production | 15–30 |
| Industrial processes | Ethylene oxide production | ~100 |
| | Petrochemical | 8–20 |
| | Refineries | ~3–13 |
| Gas combustion | Vehicles | ~12–14 |
| Extraction | Gas natural mining | ~5–70 |
| | Ambient air | ~0.04 |
| Environment | Volcanos | ~2–50 |

### 3.2. Bibliometric Analysis of Scientific Review Papers: Technical Information

#### 3.2.1. Carbon Capture

Capturing $CO_2$ constitutes the first stage of the $CO_2$ recovery process, considering both utilization (carbon capture and utilization, CCU) and storage (carbon capture and storage, CCS). Carbon dioxide is emitted in large quantities and in different concentrations by several industries and processes, including oil refineries, the cement industry, the iron and steel industry, biogas sweetening, bioindustries, and the chemicals sector [110,111], as summarized in Table 8. This information is strongly relevant to determine the most suitable technology to capture and separate the carbon.

Considering the concentrations and volumes of emitted $CO_2$, biomass and industrial processes, as well as power generation plants, are regarded as the most suitable sources of $CO_2$ for CCU [112], which may be performed through post-combustion, pre-combustion, and a combination of oxyfuel and post-combustion strategies, as discussed in the sections below. The main differences among these technologies depend on (a) how combustion is carried out, (b) how and when $CO_2$ is separated, and (c) how large the $CO_2$ concentration is in the gas stream. Consequently, the selected strategy exerts an important influence on the separation technology employed to purify the $CO_2$ stream, as this operation depends on the $CO_2$ output flowrates, $CO_2$ feed purity, and type of output stream [113]. Most of the studies and actual industrial units working commercially are based on fossil-fuel-fired power plants, when coal, oil, or gas are used for power generation. Nevertheless, it must be clear that $CO_2$ capture techniques can also be employed in combination with other source gas streams.

After $CO_2$ capture, various distinct technologies can be used for gas separation including absorption, adsorption, membrane separation, cryogenic separation, and purification through electrochemical cells [112,114]. Some of these technologies are more mature than others, but the selected option must consider mainly the $CO_2$ concentration in the source gas stream and the desired $CO_2$ purity. Moreover, the specific cost for $CO_2$ capture tends to decrease when $CO_2$ is captured with high partial pressure and in small amounts, when compared to the capture of large amounts at low $CO_2$ concentrations [115]. $CO_2$ separation processes developed for removal of $CO_2$ from other gas mixtures, including air ($N_2/O_2/CO_2$), natural gas ($CO_2/CH_4$), and $CO/CO_2$, are not mentioned.

- *$CO_2$ Source: Fuel-Fire Power and Heat Generation Stage*

Pre-combustion capture is a process where the fuel (natural gas or coal) is pretreated and then converted into syngas prior to combustion (Equations (1) and (2)). After the syngas formation, water is added to the reacting mixture to form additional amounts of hydrogen through a reaction with CO via the water gas shift (WGS) reaction process, generating a mixture of $H_2$ and $CO_2$ (Equation (3)) [105]. Due to the high concentration of $CO_2$ (>35% $CO_2$) and high pressures (30–70 atm), $CO_2$ separation is facilitated [105,116,117]. $CO_2$ is captured mainly by physical absorption, which makes use of mild commercial solvents (including Selexol™ or dimethylether of polyethylene glycol, and Rectisol™ or cold methanol), as well as membrane separation processes [118]. A hydrogen-rich fuel gas stream is also produced in the process and sent to a gas turbine to produce power [108]. As the proposed fuel conversion scheme can generate a complex combined reaction and purification process, it is normally more difficult to apply this technology to existing power plants [117,119].

$$Coal \xrightarrow{gasification} CO + H_2. \tag{1}$$

$$CH_4 + H_2O \xrightarrow{reform} CO + H_2. \tag{2}$$

$$CO + H_2O \xrightarrow{water-gas\ shift} CO_2 + H_2. \tag{3}$$

On the other hand, capturing $CO_2$ through post-combustion strategies usually constitutes the easiest path to recover $CO_2$ at power generation units, as fuel combustion is performed as usual. $CO_2$ is separated from combustion exhaust gases (flue gas) and adaptation of existing equipment is not necessary, making post-combustion the commonest industrial technology for carbon capture in actual industrial sites (85–90% of industries) [120,121]. Using this method, $CO_2$ is frequently removed from the flue gas using aqueous amine solutions at relatively low temperature (50 °C) with the help of a process known as wet scrubbing [119]. After absorbing $CO_2$, the solvent is then heated (to around 120 °C) before being cooled and recycled continuously for use in the upcoming cycle of the separation process [118]. This desorber regeneration process constitutes a very energy consuming part of the process, such that $CO_2$ is captured at a price that is essentially defined by the regeneration energetics of the amine solution [119]. For instance, retrofitting post-combustion capture at steam cracker furnaces has been estimated to cost between 37 and 70 EUR/t$CO_2$ (flue gas with 8% $CO_2$ and high concentrations of methane and hydrogen) [44]. However, since the $CO_2$ level in combustion flue gas is normally low (between 7% and 14% for coal-fired and as low as 4% for gas-fired furnaces), the energy penalty and associated costs for production of $CO_2$ streams with high concentrations (above 95.5%) can be elevated [117]. Another important challenge is the necessity to process large amounts of flue gas [119]. Studies show that the electrical power cost can increase by 65–70% for coal-fired plants and by 32% in gas plants [105] when these $CO_2$ capture strategies are employed.

According to MacDowell et al. (2010) [106], post-combustion $CO_2$ capture can be regarded as a mature technology, with several pilot and demonstration plants currently in operation across the world, managed by Alstom, Dow, PGE, E.ON, RWE Npower, Equinor, Total, Endesa, and Hitachi, among others. The principal barriers associated with the deployment of this option, apart from the economic issues, are associated with the scale-up of this technology and the inherent limitations of currently available absorption technologies.

When only oxygen is used for combustion instead of air, combustion is known as the oxyfuel combustion process and produces mainly water vapor and $CO_2$ in the exhaust gas, which can be easily separated by distillation. Thus, as pure oxygen is used, the oxyfuel process is also referred as a low-carbon-intensive combustion system, which produces fewer impurities (mainly $NO_x$) [55,108]. However, it is important to observe that (i) it can be difficult to adapt the technology to an existing plant, as heat and mass transfers and the reaction kinetics in the oxyfuel atmosphere are significantly different, resulting in flame instabilities [43], and (ii) the installation of the new plant can be expensive, due to the necessity to build an air separation unit (ASU) to produce the required oxygen [122]. The ASU is usually based on cryogenic separation or pressure swing adsorption (PSA), but the use of

molecular sieves, as well as polymeric and ceramic membranes, has also been positively evaluated, leading to a decrease in the economical and energetic costs [43,106,123]. Even so, some studies showed that $CO_2$ capture in the oxyfuel process can be more economical than in the post-combustion process [124,125] and that this advantageous aspect becomes even more important with the increase in oxygen production efficiency [43]. As an example, Johansson et al. (2012) [44] compared the post-combustion (using an amine as a solvent) and oxyfuel processes, concluding that the lower cost limits of both processes were similar, although the post-consumption process presents the advantage of being implemented more easily.

In a recent study by Amghizar et al. (2020), oxyfuel combustion was regarded as one of the most important innovative technologies for more sustainable production of olefins by steam cracking, although the authors also considered that the use of this technology is not feasible at the moment, due to the high cost and energy penalty associated with the production of pure oxygen [43].

Other promising technologies for a reduction in $CO_2$ emissions during power generation include the reduction of coke formation on the reactor inner walls to enhance heat transfer, development of novel furnace designs, design of heat recovery schemes, and use of green electricity. In fact, due to power and heat demands, energy integration can also be used to reduce costs, and green energy should be applied in combination with all the three analyzed methodologies [115], which are summarized in Table 9. In particular, all the three analyzed capture strategies have been implemented at a demonstration scale and some commercial-scale plants. Moreover, these three technologies can be fully applied in the plastics industry for generation of monomer streams and heat, although actual implementations may depend on the available infrastructure [122]. Figure 10 presents the interests and the carbon capture methods described previously in this section, and Figure 11 shows the flowcharts of the capture systems and their respective streams.

**Table 9.** Fundamental characteristics of carbon capture techniques.

| Technique/Maturity | Type of Power Plant/Plant Efficiency (%) [112] | Advantages | Drawbacks | Separation |
|---|---|---|---|---|
| Pre-combustion [113,128] <br> - TRL 8 (considering solid sorbents and absorption) [123]; <br> - Up to TRL 5 (considering membranes) [123] | - IGCC [a]/35 | - Separation of $CO_2$ is easier due to high partial pressure; <br> - Combustion only with $H_2$; <br> - High maturity (commercially deployed at large scale); <br> - Opportunity to retrofit in existing plants; <br> - Ideal for IGCC plants; <br> - The energy requirements may be half than for post-combustion capture. | - Heat transfer in the turbine has to be adjusted; <br> - Efficiency penalty due to WGS reaction when using $CH_4$. | - PSA [118]; <br> - Membranes [112]; <br> - Absorption by solid sorbents (porous organic frameworks and membranes); <br> - Absorption by physical solvents (Selexol[TM], Rectisol[TM]); <br> - Adsorption by chemical solvents (amine-based solvent, typically MEA); <br> - Gas hydrate [112]; <br> - Cryogenic processes [112]. |

**Table 9.** *Cont.*

| Technique/Maturity | Type of Power Plant/Plant Efficiency (%) [112] | Advantages | Drawbacks | Separation |
|---|---|---|---|---|
| Post-combustion [105,119,128,129] <br> - TRL 8 (considering absorption) [123] <br> - TRL 6 (considering solid sorbents and CLC) [123] <br> - Up to TRL 5 (considering membranes) [123] | - NGCC [b]/49 <br> - PC [c]/31 | - Capture from streams with low $CO_2$ partial pressure; <br> - Mature technology [106]; <br> - easy to be implemented as retrofits more easily; <br> - Commercially deployed at large scale. | - Requires high volumes of gas; <br> - High energy demand; <br> - Costly (increase costs in over 30%) <br> - Scale-up [106] | - Adsorption by chemical solvents (amine-based solvents, including MEA, DEA, KS-1; alkaline solvents, including NaOH and $Ca(OH)_2$; ionic liquids) <br> - Absorption by solid sorbents (amine-based solid sorbents; alkali earth metal-based solid sorbents, such as $CaCO_3$; alkali metal carbonate solid sorbents, such as $Na_2CO_3$ and $K_2CO_3$; porous organic frame-works/polymers) <br> - Membrane separation (polymeric membranes; inorganic membranes, typically zeolites; hybrid membranes); <br> - Cryogenic separation; <br> - Pressure/vacuum adsorption (zeolites; activated carbon). |
| Oxyfuel [43,44,97,100] <br> - TRL 7 (considering cryogenics) [123] <br> - TRL 6 (considering solid sorbents and CLC) [123] <br> - Up to TRL 5 (considering membranes) [123] | - NGCC [b]/46 <br> - PC [c]/33 | - Requires smaller equipment; <br> - lower energy demand; <br> - Reduction in NOx formation; <br> - Generation of a high-purity $CO_2$ stream. | - Air separation unit is costly; <br> - High energy demand; <br> - Flame instabilities; <br> - Lower efficiency. | - $CO_2$ is separated by condensing water vapor; <br> - nitrogen is separated from air by cryogenic or membrane technologies; <br> - chemical looping combustion (CLC); <br> - chemical looping reforming. |

[a] IGCC: integrated gasification combined cycle; [b] NGCC: natural gas combined cycle; [c] PC: pulverized coal.; TRL: technology readiness level; MEA: monoethanolamine; DEA: diethanolamine; KS-1: proprietary solvent developed by Mitsubichi[TM]

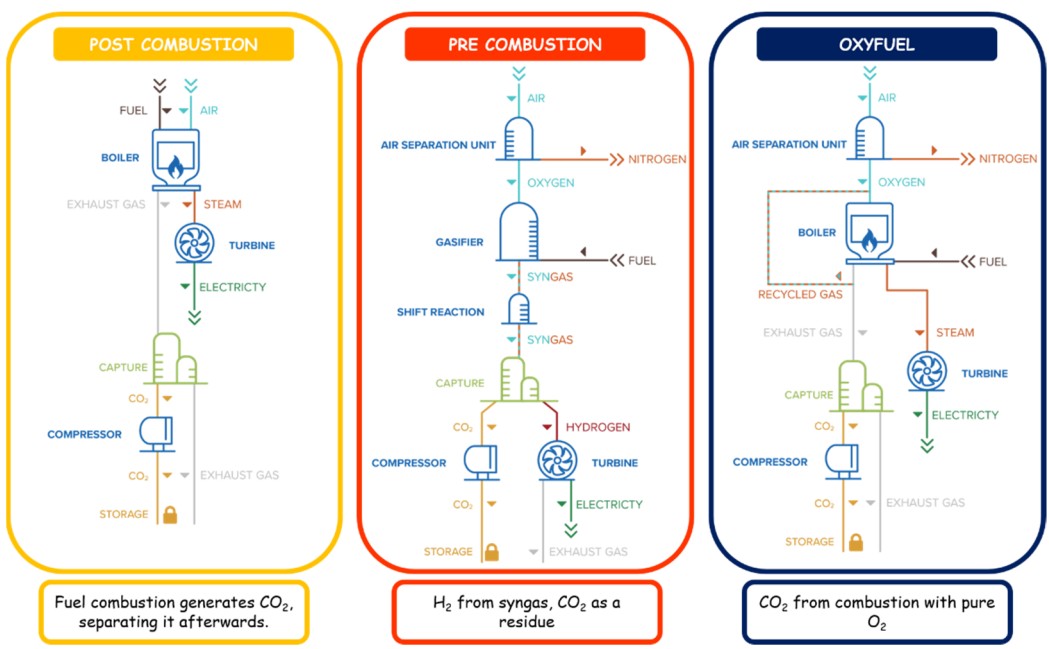

**Figure 10.** Techniques for capture of carbon dioxide during power generation (provided by Global carbon capture and storage (CCS) Institute). Source: [120,126].

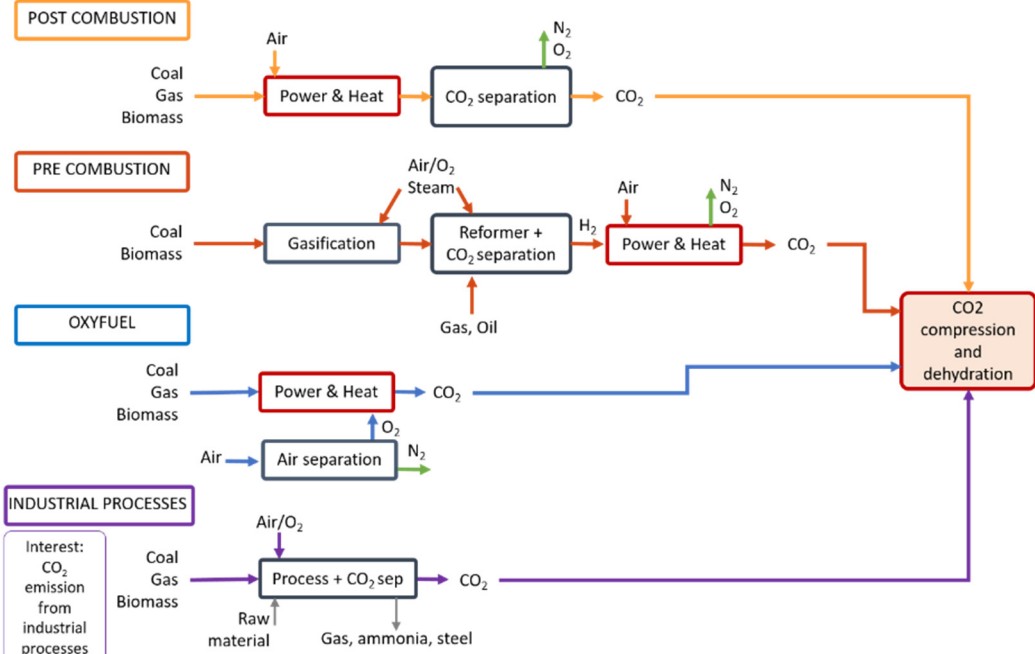

**Figure 11.** Flowchart of $CO_2$ capture processes and systems. Adapted from [127].

As in the case of the oxyfuel process, other modern technologies are being developed to simultaneously capture and enrich the $CO_2$ stream. For instance, chemical looping combustion (CLC) or chemical looping reforming (CLR) uses a solid sorbent to absorb oxygen (frequently a metal oxide, named the oxygen carrier), which is afterward put in contact with a fuel, promoting the combustion/reforming reactions [122,129–131]. The resulting exhaust stream contains only carbon dioxide and water vapor (as in the oxyfuel combustion process), as illustrated in Figure 12. The CLC technique has the potential to be the cheapest carbon capture system, although it is still immature, facing challenges related to selection of the most appropriate oxygen carrier and handling of process materi-

als [122,129]. Particularly, oxygen carriers must also be developed for CLR, and the overall technology is still at the laboratory or concept stage of development, with few pilot-scale studies currently under investigation [114].

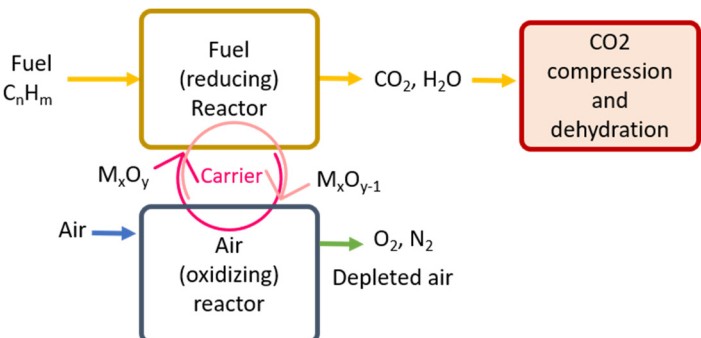

**Figure 12.** Scheme and streams of the CLC process. Adapted from: [131].

Fuel cell systems (particularly solid oxide fuel cells, SOFCs) are electrochemical devices that can also be used to simultaneously produce power and heat. As an inherent characteristic, air and fuel streams are kept separate, facilitating the final capture of carbon [132]. Moreover, advanced supercritical $CO_2$ combustion processes that employ recycled $CO_2$ and operate over the supercritical pressure of $CO_2$ have been regarded as promising technologies for utilization of emitted $CO_2$. The use of supercritical $CO_2$ as a working fluid in power cycles has been considered in distinct scenarios and can apparently lead to development of more energy-efficient plants, when compared to the conventional steam cycle. Particularly, it has been shown that the steam turbine efficiency can be increased if supercritical $CO_2$ is used as the working fluid. The main disadvantage of this process is the need to purify the oxygen stream before use [114].

It must also be said that there are ongoing research attempts to develop and commercialize algae-based carbon conversion technologies. In addition to enabling carbon capture by biofixation in high amounts (one kilogram of dry algal cells fix 1.83 kg of $CO_2$), companies are developing technologies for useful conversion of the sequestered carbon [133,134]. In addition, the use of algae-based $CO_2$ mitigation strategies can generate many valuable products that can be used to generate significant revenues, thereby making this route technically and economically feasible [122,135,136]. However, considering the scope of the present work and the recommended operation conditions, the technology has not been recommended so far as a feasible alternative for the plastics industry.

Figure 13 presents the distribution of the main carbon sources of the analyzed reviews. It is interesting to note that most papers do not specify the source of the $CO_2$, focusing on technologies that can be employed to enrich and use the carbon stream after its capture. Despite that, industrial processes are the most cited sources of $CO_2$, corresponding to almost 80% of the total available references. In this case, power plants, the cement industry, refinery waste gases, biological treatment [137], and combustion were the most cited industrial processes. For instance, the four most cited review papers of Table 7 consider industrial processes as viable carbon sources for capture and posterior storage or use of carbon dioxide. This relevant concentration of references about industrial processes may in fact reflect the growing concern of process industries with environmental pollution and commercial image. "Others" in Figure 13 correspond to gas purification process streams (natural gas [138] and biogas [139]), preparation of synthetic gaseous mixtures [140] and residual agroforestry process streams [141]. Capture of $CO_2$ from the atmosphere was mentioned by 8% of the review papers that specified the analyzed carbon sources [142–154].

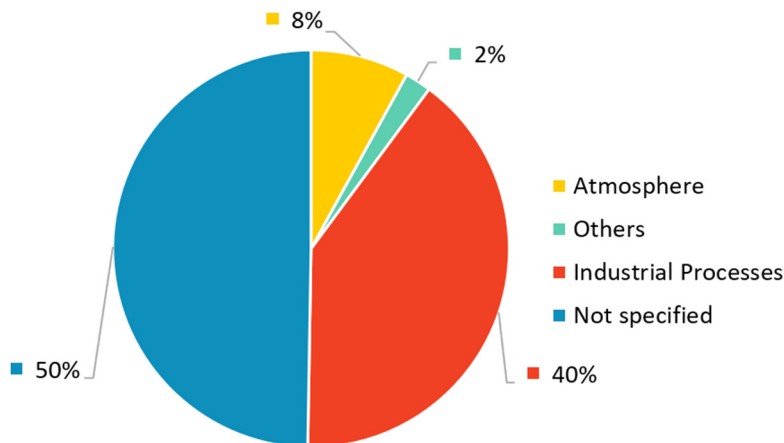

**Figure 13.** Most cited $CO_2$ sources in review papers.

3.2.2. $CO_2$ Separation Techniques

Following the carbon capture process, the concentration of $CO_2$ in the gas stream must almost always be enriched for posterior conversion or storage. The complexity of this process step depends on the composition of the gas mixture and the $CO_2$ concentration. As an example, the simple gas liquefaction operation can be performed, as in the case of the Sabic ethylene oxide production [155]. In this case, carbon dioxide and water are generated simultaneously due to existing side reactions, although $CO_2$-rich gas streams can be easily obtained through simple water condensation (Table 10).

**Table 10.** Kinetic mechanism of usual ethylene oxide manufacture [155].

| Reaction | Stoichiometry |
|---|---|
| Main ethylene oxide reaction | $C_2H_4 + 0.5\,O_2 \overset{Ag}{\rightarrow} C_2H_4O$ |
| Undesired side reactions | $C_2H_4 \rightarrow CH_3CHO + \frac{5}{2}\,O_2 \rightarrow 2H_2O + 2CO_2$ |
| | $C_2H_4 + \frac{3}{2}\,O_2 \overset{Ag}{\rightarrow} 2H_2O + 2CO_2$ |

However, when the $CO_2$ concentration is small and/or the gas stream is more complex and contains a large number of chemical components, other separation techniques must be considered for $CO_2$ purification. These techniques are usually known for their characteristic high costs and high energy demands (for instance, an additional energy penalty of 20% to 30% can be observed in a power generation system [29]). Fortunately, these are mature technologies that are characterized by high TRL values, although some of them are still subject to accelerated development and being investigated by many companies, reducing the risks of use and the possible drawbacks associated with the implementation of poorly known technologies.

It is not intended to describe all possible $CO_2$ separation technologies in detail in this section, but to provide an overview of the available processes and discuss the applicability of each process in the plastics industry (Tables 11 and 12, and Figures 14 and 15). Several reviews are available elsewhere for presentation of additional details about the analyzed $CO_2$ separation technologies [108,114,118,122,129,156–159]. Figure 14 shows the most relevant $CO_2$ purification techniques, as described in the literature. It must be clear that this is a field where new technologies are under development and where the number of reviews is increasing fast due to the urgency of GHG mitigation. In particular, it can be said that chemical absorption still is the most studied $CO_2$ purification technique, followed by adsorption and membrane separation processes. As already said, these three technologies are robust, mature, and suitable for processing of flue gas streams with distinct characteristics and production of $CO_2$ streams with distinct $CO_2$ concentration levels, as discussed in this section.

**Table 11.** Most relevant $CO_2$ purification technologies.

| Separation Technique | Definition |
|---|---|
| Absorption | It can be classified as chemical or physical. In the first case, a chemical reaction occurs between an absorbent molecule and $CO_2$. In the second case, the physical solvent readily incorporates and dissolves $CO_2$, and weak binding forces between gas and solvent molecules keep $CO_2$ in solution. Regeneration is frequently performed through pressure swing [129,160]. |
| Adsorption | A porous material with high specific surface area is used for adsorption of $CO_2$ molecules through intermolecular forces. Selective adsorption of the gases depends on temperature, partial pressures, surface forces, and adsorbent pore size distribution. Adsorbent beds are regenerated mainly by pressure swing and temperature swing methods [160–162]. |
| Membranes | Semi-permeable barriers prepared with different materials (organic or inorganic) can be used to separate substances through various mechanisms, including solution/diffusion, adsorption/diffusion, molecular sieving, ionic transport, and facilitated transport [163]. |
| Cryogenic separation process | Distillation of $CO_2$ can be performed at low temperatures and high pressures [161]. |
| Hybrid processes | Two or more separation subsystems can be combined to improve separation efficiency and reduce costs, as in the case of combined adsorption/membrane systems [114,161]. |
| Hydrate separation | Hydrates are initially formed by exposing the $CO_2$-rich exhaust gas stream to water under high pressure. As hydrates are formed, $CO_2$ is captured. Hydrates are then separated and dissociated, releasing $CO_2$ in pure form [164]. |
| Biofixation | Autotrophic microorganisms fix $CO_2$ to synthesize biological organic materials and extracellular products, capturing $CO_2$ from the atmosphere or other sources (including flue gases) with help of different cultivation systems. As a result, $CO_2$ is fixed in the form of organic biomass, which can be converted into different chemicals and biofuels through biorefining processes [165,166]. |
| Mineralization | Mineralization involves the reaction of $CO_2$ with materials that contain alkaline earth oxides such as magnesium oxide (MgO) and calcium oxide (CaO). Valuable carbonate products can be obtained from industrial byproducts and wastes [167]. |
| Calcium looping processes (CLP) | The CLP technology is a special mineralization technique that removes $CO_2$ from flue gas by carbonating calcium oxide (CaO) to calcium carbonate ($CaCO_3$) [168]. |
| Molten carbonate fuel cell (MCFC) | The MCFC technology uses a molten carbonate salt suspended in a porous ceramic matrix as an electrolyte that is able to perform the internal reforming reaction, converting other fuels to hydrogen [169]. |
| Electrochemically mediated amine regeneration (EMAR) | The EMAR technology constitutes an alternative route for regeneration of amines for carbon capture from a flue gas source that does not make use of thermal regeneration, saving energy and increasing the $CO_2$ capture efficiency [170]. |

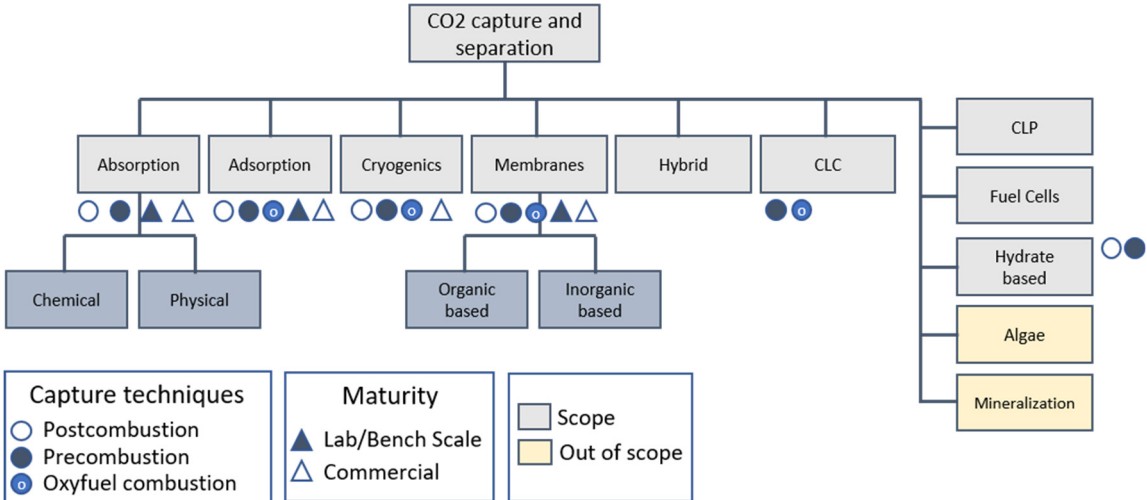

**Figure 14.** Relationship between carbon capture and separation techniques. Data source: [112,114].

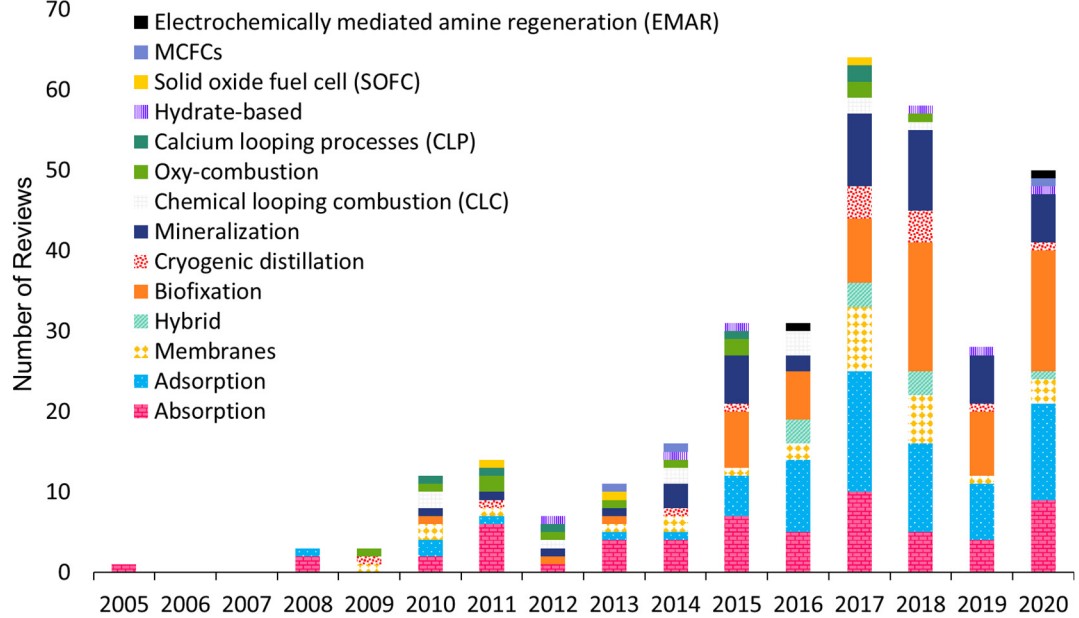

**Figure 15.** Annual citation frequency distribution of $CO_2$ capture and separation techniques in review papers.

Some characteristics of oxyfuel combustion, chemical looping combustion, and fuel cell processes were presented in the previous section, as they can be used simultaneously to capture and to enrich the $CO_2$ stream. Mineralization technologies are considered suitable for both capture and utilization of $CO_2$, storing carbon in a solid form, which is very convenient for environmental purposes. However, this technique is not discussed in detail because the formation of mineral carbonates (Mg or Ca) is not very interesting for the plastics industry (although carbonates can be used as additives during compounding of commercial plastic grades [171]). Biofixation by algae is not interesting for the plastics industry either if the biomass is used for the manufacture of biofuels (although use of biofuels at plant site can reduce the environmental impact of heat generation in the plastics industry [172,173]). The most valuable separation techniques are shown in Figure 14, with their respective levels of maturity (lab or commercial scale) and the $CO_2$ capture techniques that are commonly associated with the analyzed separation strategy.

- *Absorption*

**Table 12.** Citing frequency of $CO_2$ capture and separation techniques.

| Capture and Separation Processes | No. of Identified Processes | Percentage (%) | References |
|---|---|---|---|
| Absorption | 84 | 21.7 | [73,104–108,112,118,119,124,126,138,144,156,157,159,174–212] |
| Adsorption | 79 | 20.4 | [73,105,110–112,138,139,141,143,145,153,156,157,175,180,187,188,190,194,195,197,199,200,202,208,213–252] |
| Biofixation/bioconversion | 66 | 17.1 | [95,126,137,142,145,146,197,253–301] |
| Mineralization | 48 | 12.4 | [95,105,116,117,126,142,150,151,154,157,158,180,187,273,284,285,293,298,302–324] |
| Membranes | 32 | 8.3 | [73,105,106,110,112,118,119,124,138,152,156,157,175,187,193,196,197,199,202,235,325–330] |
| Hybrid | 21 | 5.4 | [114,156,157,193,202,220,262,331,332] |
| Cryogenic distillation | 14 | 3.6 | [105,108,110,112,138,156,157,187,196,197,204,325,330] |
| Oxyfuel combustion | 12 | 3.1 | [105,106,112,118,124,157,188,333–337] |
| Chemical looping combustion (CLC) | 11 | 2.8 | [73,105,106,157,196,197,216,298,326,338–340] |
| Calcium looping processes (CLP) | 6 | 1.6 | [117,157,196,341–343] |
| Hydrate-based | 6 | 1.6 | [105,112,164,187,197,344] |
| Solid oxide fuel cell (SOFC) | 3 | 0.8 | [124,196,345] |
| Molten carbonate fuel cell (MCFC) | 3 | 0.8 | [124,326,346] |
| Electrochemically mediated amine regeneration (EMAR) | 2 | 0.5 | [298,346] |

Absorption with chemical solvents is the most mature and most employed $CO_2$ separation technology, especially using amines [128,161]. The main advantage of these processes is the capacity to capture $CO_2$ from flue gases with low $CO_2$ partial pressures, which is the case of most $CO_2$ streams available in the plastics industry. The high technological development is reflected in the high number of publications that provide reviews of this technology (see Table 12). Particularly, Figure 16 presents the reaction mechanism between $CO_2$ and monoethanolamine (MEA)/diethanolamine (DEA), some of the commonest absorbing solvents [108], while Figure 17 shows a simplified schematic representation of the amine-based $CO_2$ capture from flue gas.

**Figure 16.** Capture of $CO_2$ by MEA or DEA, forming carbamate and bicarbonate, respectively.

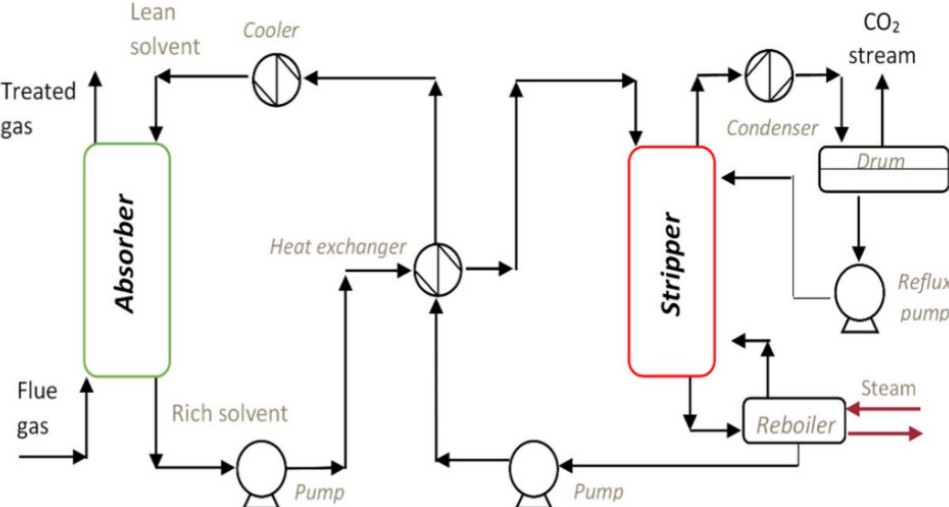

**Figure 17.** Simplified schematic representation of the amine-based $CO_2$ capture and separation from flue gas. Reproduced with permission from Álvaro A. Ramírez-Santos, Christophe Castel, Eric Favre, Separation and Purification Technology; published by Elsevier, 2018 [156].

Despite its popularity, chemical absorption is known to demand high amounts of energy. Furthermore, retrofitting of a chemical absorption plant based on amines can be hard due to the thermal demands associated with the stripping section (Figure 17) [347], which may not be financially sustainable. Another important challenge is associated with the oxidative degradation of the solvent due to the presence of oxygen in the flue gas [182]. To overcome these issues, the use of new solvents is being investigated, including piperazine (PZ) and its derivatives, chilled and aqueous ammonia, and ionic liquids (ILs) [114,176]. These new solvents are also improving the $CO_2$ loading capacity and, due to these innovations, chemical absorption remains the most developed carbon dioxide separation technology and still constitutes a very active field of research. For example, the use of aqueous ammonia (wet scrubbing) for $CO_2$ capture and purification can lead to a lower heat of reaction (saving 60% energy when compared to MEA), increase the $CO_2$ absorption capacity, allow regeneration at high pressure (saving energy consumption for pressure transitions), be more tolerant to oxygen, and be more stable and less prone to degradation [184,201]. However, equipment plugging due to formation of solids and significant loss of ammonia vapor during stripping can occur [110,118]. Interesting discussions about the commercial deployment and development of new absorbing solvents are available elsewhere [184,196,201,348].

Physical absorption uses a physical solvent that can readily dissolve $CO_2$ and does not require the use of heat for regeneration. However, the technique demands the use of low-temperature, pressurized, and $CO_2$-rich gaseous streams [129], which is more compatible with the pre-combustion capture scheme. In the proposed analysis, 21.5% out of the 79 absorption citations described in Table 12 were related to physical absorption, which mentioned the use of commercial solvents: Selexol (dimethyl ethers of polyethylene glycol, DMPEG), Rectisol (methanol), and Purisol (*N*-methyl-2-pyrolidone, NMP).

- *Adsorption*

Adsorption-based systems are the second most cited carbon capture and separation technology in the review papers. They include pressure swing adsorption (PSA), vacuum swing adsorption (VSA), and temperature swing adsorption (TSA). In this system, $CO_2$ is physically or chemically adsorbed onto adsorbent sites and/or dissolves into the pore structure of the solid. The use of high $CO_2$ partial pressures enhances the adsorption rates and capacities, making these sorbents more useful for high-pressure applications [129]. While PSA certainly presents a high level of maturity due to its frequent use in pre-combustion systems, VSA and TSA technologies can be better applied in post-combustion

$CO_2$ capture problems, although the levels of maturity of VSA and TSA are small and these technologies are still far from large-scale implementation [114,122].

- *Biofixation*

   Biofixation is the third most mentioned $CO_2$ capture process of Table 12, being mentioned in 17.1% of the total process citations. This biological process occurs mainly through acidophilic microalgae and is commercially available. The main advantage of biofixation is the production of lipids, which can be further converted into ethanol and used to produce green monomers, such as ethylene, through known commercial reaction routes. Therefore, the main interest of $CO_2$ biofixation for the plastics industry can be associated with biomass bioconversion. However, although biofixation can be used to sequester $CO_2$ from combustion flue gas, enormous land areas and water volumes may be required [264]. In addition, the flue gas has to be purified to remove pollutants (such as $SO_x$, $NO_x$, and heavy metals) that are toxic for the microalgae [268]. Temperature (should not be higher than 45 °C), pH, light intensity and nutrient concentrations (carbon, nitrogen and phosphorus) must also be tightly controlled [268,298]. Thus, additional studies based on the use of flue gas as carbon source are required to investigate the effect of flue gas composition on $CO_2$ biofixation. If suitable, the best configuration may eventually make use of multi-layer bioreactors due to easiness to scale-up, lower energy consumption, lower required installation areas and higher rates of cell growth [268].

- *Membrane separations*

   The use of membranes for $CO_2$ capture and separation was mentioned at least in 9% of the total process citations, emphasizing the fast development of membrane materials and the economic advantages of these processes, especially when it is necessary to achieve high $CO_2$ recovery and purity simultaneously [161,349,350]. Moreover, the use of composite membranes manufactured with polymeric and inorganic constituents may constitute the future trend for membrane-based $CO_2$ separation processes [114]. However, these processes usually require the compression of the gas stream, which can be sometimes difficult and uneconomic. For this reason, $CO_2$ membrane separations are better suited for purification of high-pressure streams with high $CO_2$ concentrations. In addition, membrane-based $CO_2$ separations often require the implementation of multi-stage operations, stream recycling and cooling of flue gases (below 100 °C) to prevent the membrane degradation [105,118,156], leading to additional operation costs.

- *Hybrid technologies*

   As shown in Table 12, 6.81% of the total process citations are related to the use of hybrid technologies, when two or more separation units or systems are combined in parallel, in series, or in a single piece of equipment to enhance the separation efficiency and reduce the overall cost of the separation [114]. Hybrid membrane–PSA systems, hybrid pressure–temperature swing adsorption processes (PTSA), combined membrane separation–cryogenic distillation processes, and combined membrane–absorption separations constitute some of the analyzed examples [114,331,351]. Very frequently, the use of membrane separations in the first separation stage for enrichment of the $CO_2$ concentration and posterior application of absorption, PSA or cryogenic distillation can lead to more economical and compact separation units [352]. Although the use of hybrid technologies has been investigated less frequently, available studies have highlighted that hybrid technologies can be very useful in actual industrial applications due to the enhanced energy efficiency and cost effectiveness [114,331].

- *Cryogenic separation*

   Some reviews reported the use of cryogenic separations for post-combustion $CO_2$ capture applications, although the high energy demand of this process makes this solution uneconomic in most cases [162]. For this reason, this technology is normally suitable for streams that contain high concentrations of $CO_2$ (usually more than 50 mol%) and is usually unfeasible for large commercial deployment, being mentioned in 3.94% of the total process citations [162,331]. The absence of a sorbent and the operation at atmospheric pressure are

some of the process advantages. Besides, water can also be removed without an additional process step, although water must be carefully removed from process vessels and lines because of the frequent formation of ice particles and plugging of vessels and ducts.

- *Other separation techniques*

Chemical absorption catalyzed by the enzyme carbonic anhydrase (CA) can possibly be regarded as a hybrid technology. As a matter of fact, CA converts $CO_2$ directly to bicarbonate ($HCO_3^-$) at high turnover rates, slightly reducing the heat of desorption and improving the overall efficiency of solvents or membranes (enzymatic membrane contactors) [332]. However, the sensitivity to temperature and impurities and limited lifetime of the enzyme (roughly 6 months) are some important process drawbacks [193]. Thus, enzyme-based $CO_2$ capture constitutes a viable and promising new technology, although additional studies are required to investigate and enhance the enzyme activity and long-term stability [193,332].

It must be observed that, although the calcium looping process (CLP) is not a new technology (dates to 1867), CLP was mentioned in only 2.15% of the total process citations. Liu et al. (2012) stated that "*the renewed interest in the CLP is because of the abundant reserves of potential sorbents and high efficiency or low cost associated with the process*" [342]. The system is similar to CLC, although, in the CLP process, calcium oxide (CaO) is repeatedly cycled between two vessels. The carbonation of CaO occurs in the carbonator, removing $CO_2$ from the flue gas and forming $CaCO_3$, which is transferred to another vessel where calcination takes place in the calciner, CaO is formed and finally transferred back to the carbonator, leaving a pure stream of $CO_2$ behind [341,343]. However, some practical problems inhibit most industrial applications, including the fact that the $CO_2$ capture capacity decreases during the sequential carbonation/calcination cycles. A summary of the main process routes that involve CLP was presented by Abanades et al. (2015) [123].

An interesting point is the fact that the separation of $CO_2$ can be made possible through application of electrical potential gradients in electrochemical cells [129,353–356]. Electrochemistry is used in this case for electrochemical reduction of $CO_2$ using an electric current. However, for the electrochemical transformation of $CO_2$ in electrochemical cells, active and selective electrocatalysts are also necessary, although the lack of efficient and stable catalysts constitutes one of the main bottlenecks for the development of $CO_2$ electroreduction processes [357]. Other problems can be associated with gas diffusion electrode technology. A possible solution may involve the use of ionic liquids (similar to those used in absorption systems) due to their inherent electrical conductivity [356,357]. Nevertheless, although electrochemistry can provide a path for in situ generation of synthetic fuels or chemicals directly from $CO_2$, few published materials discuss its use for both capture and utilization of $CO_2$ [336,356,357], resulting in less than 1% of the total process citations. An example of an electrochemical method used for CCU purposes is the reverse mode of a molten carbonate fuel cell (*MCFC*) [132,354–356,358–360]. In MCFCs, the fuel (natural gas or biogas) is oxidized, generating energy, while CO is simultaneously moved from the cathode to the anode, facilitating the separation by condensation [124,326]. MCFCs are known to be efficient and to make cogeneration possible; nonetheless, the requirement of large power FC installations and its high cost are major drawbacks of this technology [65,326,360]. Lee (2014) wrote a review that is focused specifically on MCFCs [326].

Electrochemically mediated amine regeneration (EMAR) is a process that can be regarded as a hybrid system, where $CO_2$ is captured by chemical absorption using amines, although the $CO_2$ desorption is performed electrochemically at room temperature (instead of using the traditional TSP process) and at elevated pressure. Consequently, the most significant innovation in this electrically driven post-combustion system is the new regeneration methodology, which reduces absorber size and can lead to higher energetic and recovery efficiencies [347].

In conclusion, many different technologies have been developed and used for $CO_2$ capture and separation. In order to capture flue gases from combustion, cracking, and polymerization process emissions, focuses of future technologies should lie on better solvents,

faster $CO_2$ transport kinetics, less energy-demanding desorption steps, optimization of the heat required to perform regeneration of solid sorbents and solvents, faster cooling of process streams, and new membranes that can provide better tradeoffs between recovery rates and product purities and higher selectivities [114,129,160]. Even so, the present scenario may change in the coming decade due to evolving breakthroughs and significant increase of in number of studies and patents regarding $CO_2$ capture and separation.

- *Additional comments*

As mentioned before, the $CO_2$ concentrations are small in most carbon sources found in the plastics industry (5–25% of $CO_2$) [44,361]. For this reason, most CCU technologies are not well suited for the carbon dioxide concentrations available in the plastics industry (physical absorption, PSA), are not mature enough (including electrochemistry, adsorption, membrane, TSA, and CLC processes), or may be difficult to retrofit to existing facilities (such as biofixation and cryogenic separations). Under the current technological stage, only chemical absorption seems capable of capturing $CO_2$ from gas streams available in typical plastic plants, which contain low $CO_2$ concentrations, to provide streams with high $CO_2$ purity. Membrane and adsorption separation processes might be further used for $CO_2$ capture from gas streams with low $CO_2$ concentrations as the current level of development does not support the present applications [44]. In addition, TSA and membrane purification processes produce output streams with lower $CO_2$ purity than chemical absorption, which can constitute a hurdle for posterior conversion of $CO_2$ [347,362,363].

Contrary to the low $CO_2$ partial pressures of flue gases emitted by typical plastics industries, the production of ethylene oxide (needed for PET, the fourth most demanded plastic [23]) generates a $CO_2$-rich gas stream and can be regarded as an exception [159,364]. Ethylene oxide is currently produced by the catalytic partial oxidation of ethylene with oxygen. In this case, physical absorption or adsorption constitute the best options for $CO_2$ capture [113].

### 3.3. Bibliometric Analysis of Patents Literature and Technological Players

Considering the searched documents, patent analyses were performed to identify the main players (mainly startups) involved with carbon dioxide capture, separation, and use, focusing on applications intended for plastic producers. According to the initial screening, the activities of many startups were related to mineralization, which can indeed allow the efficient capture of $CO_2$, as described in the previous section, but does not provide interesting products for plastic manufacturers, as the main idea behind mineralization is the storage of carbon in solid mineral form. It is interesting to observe, however, that plastics can also be used similarly for storage of carbon in the solid state. Therefore, in a certain sense, mineralization and plastic production can be regarded as competitors in the field of carbon capture and storage. Some of the companies that perform mineralization as a strategy for capture of $CO_2$ are Algatec, Blue Planet, CalciTech, Calera, Carbicrete, Carbon Capture Machine, Carbon Upcycling Technologies Inc., CarbFix, $CO_2$Concrete, CarbonCure, Carbonfree Chemicals (Skyonic Corporation), Cool Planet Energy Systems, EcoEra, Green Minerals, Just BioFiber Structural Solutions Corp., MOF Technologies, Mosaic Materials, Skyonic, and Tandem Technical.

On the other hand, the main technological activities of several startups can be associated with the biological capture of $CO_2$ and posterior bioconversion of the biomass into useful chemicals, which is a much more interesting $CO_2$ capture route for plastic manufacturers, despite some drawbacks discussed in the previous section. As a matter of fact, if one considers that ethanol is the main product obtained from biomass conversion, a considerably large range of bioproducts can be posteriorly manufactured, as illustrated in Figure 18. Some of the companies that perform industrial activities in this field are Algenol Biofuels, A2BE Carbon Capture, Algae AquaCulture Technology, Aljadix, Bodega Algae, EnobraQ, Grow Energy, LanzaTech, PHY, Phytonix, Pond Technologies, Phycal, SkyTree, Synthetic Genomics, and Trelys. Among these many interesting companies, it is possible to highlight the activities performed by LanzaTech (partner of Novo Holdings) (Table 13)

and Synthetic Genomics (a partner of ExxonMobil) (Table 14), which make use of bacteria (such as *Clostridium autoethanogenum*) and algae, respectively, to capture $CO_2$ from $CO_2$-rich streams and whose technologies are already available for large-scale commercial facilities. Particularly, flue gas can be used in these cases to produce biofuels such as ethanol, 1,4-butanediol, 2,3-butanediol, and isobutene [365–368]. One must consider that ethylene manufacture from ethanol constitutes a mature and well-developed technology. It must also be highlighted that, although the Algenol Biofuels technology is not fully developed for large-scale commercial facilities, this company has already deposited at least 10 patents that propose the use of cyanobacteria to produce ethanol in closed photobioreactors, as described in Table 15, with collaborations with Pacific Northwest National Laboratory (PNNL), National Renewable Energy Laboratory, and Georgia Tech [369].

**Table 13.** Patents related to bioconversion and deposited by LanzaTech.

| Code | Priority Year | Status | Name | Technology | Product | Ref. |
|---|---|---|---|---|---|---|
| WO2007117157A1 | 2006 | Application (2007) Grant in USA | Microbial fermentation of gaseous substrates to produce alcohols | Bioconversion | Alcohols | [370] |
| WO2015058011A1 | 2014 | Application (2015) Grant in CA | Carbon capture in fermentation | Bioconversion | Ethanol, acetate, and/or 2,3-butanediol | [371] |

CA: Canada.

**Table 14.** Patents related to bioconversion and deposited by Synthetic Genomics, Inc.

| Code | Priority Year | Status | Name | Technology | Product | Ref. |
|---|---|---|---|---|---|---|
| WO2019133726A1 | 2018 | Application | Genetic modulation of photosynthetic organisms for improved growth | Bioconversion | Biofuel | [372] |
| WO2017095960A1 | 2016 | Application Grant in USA | Compositions and methods for expressing genes in algae | Bioconversion | Biofuel | [373] |
| WO2017070404A2 | 2016 | Application Grant in USA | Enhanced productivity by attenuation of chlorophyll binding protein genes | Bioconversion | Biofuel | [374] |
| WO2017041048A1 | 2016 | Application Grant in USA | Microorganisms engineered for increased productivity | Bioconversion | Biofuel | [375] |
| WO2017011707A1 | 2016 | Application Grant in USA | Microorganisms having increased lipid productivity | Bioconversion | Biofuel | [375] |
| WO2015103307A1 | 2014 | Application Grant in USA and EP. | Biomass productivity regulator | Bioconversion | Biofuel | [376] |
| WO2015051342A2 | 2014 | Application Grant in USA | Compositions and methods for modulating biomass productivity | Bioconversion | Biofuel | [377] |
| WO2009098089A2 | 2009 | Application Grant in USA | Genetically modified cyanobacteria for the production of ethanol | Bioconversion | Biofuel | [378] |

EP: European patent.

**Table 15.** Patents related to bioconversion and deposited by Algenol Biofuels.

| Code | Priority Year | Status | Name | Technology | Ref. |
|---|---|---|---|---|---|
| US7682821B2 | 2006 | Grant (2010) | Closed photobioreactor system for continued daily in situ production, separation, collection, and removal of ethanol from genetically enhanced photosynthetic organisms | Conversion by cyanobacteria | [379] |
| US8691538B1 | 2012 | Grant (2014) | Biofilm photobioreactor system and method of use | Conversion by microorganisms | [380] |
| US9896652B2 | 2014 | Grant (2018) | Photobioreactor, system and method of use | Conversion by microorganisms | [381] |
| US89121012B2 | 2013 | Grant (2015) | Staged inoculation of multiple cyanobacterial photobioreactors | Conversion by cyanobacteria | [382] |
| WO2014145185A1 | 2008 | Application (2010) | Process for inoculating closed photobioreactors with cyanobacteria | Conversion by cyanobacteria | [383] |
| US8846369B2 | 2012 | Grant (2014) | Cyanobacterium sp. host cell and vector for production of chemical compounds in cyanobacterial cultures | Conversion by cyanobacteria | [384] |
| WO2014100799A3 | 2012 | Application (2014) Grant in USA and EP | Cyanobacterium sp. for production of compounds | Conversion by cyanobacteria | [385] |
| WO2007084477A1 | 2006 | Application (2007) Grant in USA, EP, JP, CA, DE, and ES | Methods and compositions for ethanol producing cyanobacteria | Conversion by cyanobacteria | [386] |
| WO2011072122A1 | 2009 | Application (2011) Grant in USA | Water/carbonate stripping for $CO_2$ capture adsorber regeneration and $CO_2$ delivery to photoautotrophs | Air capture and separation by adsorption | [387] |

EP: European patent; JP: Japan; CA: Canada; DE: Germany; ES: Spain.

Figure 19 and Table 16 present the main companies that currently invest in the development of $CO_2$ capture technologies. As also observed previously, chemical absorption is by far the commonest subject of the downloaded patents, reinforcing the many advantageous aspects of these processes described before. While amines are the commonest solvents used for $CO_2$ capture through well-established chemical absorption processes, novel types of solvents are under development and the use of enzymes can be regarded as very promising. This also helps to explain why chemical absorption technology has been applied so frequently worldwide for $CO_2$ capture, as it is still far from reaching the final limiting technological plateau. Adsorption and membrane separation processes appeared in the second and third positions, respectively, of the most frequently used technologies for $CO_2$ capture and purification, as also observed during the analysis of the scientific literature, indicating the robustness of the present discussion. This technological pattern has also been described by Li et al. (2013) [160]. Patents regarding the use of cryogenic separations, fuel cells, oxycombustion processes, and hybrid systems for $CO_2$ capture and purification were also found in the analyzed patent literature.

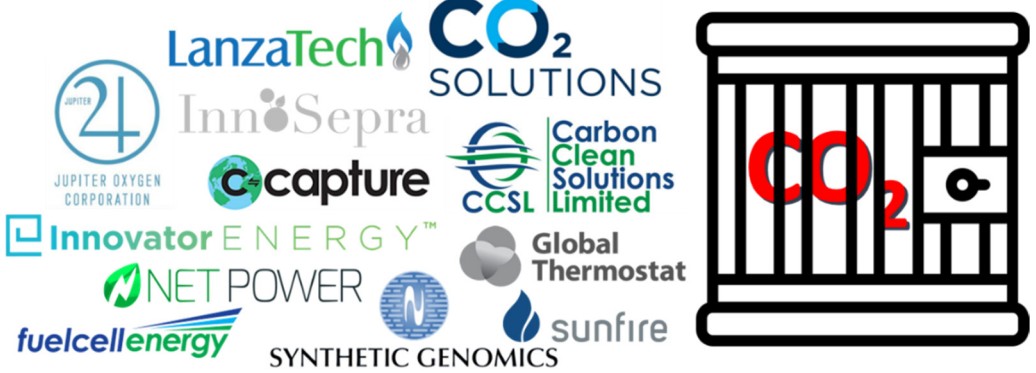

**Figure 18.** Chemical pathways for manufacture of alkene monomers and polymers from ethanol. Reproduced with permission from Robert T. Mathers, Journal of Polymer Science Part A: Polymer Chemistry; published by John Wiley and Sons, 2012 [388].

**Figure 19.** Main companies involved with the development of technologies for $CO_2$ capture and purification.

**Table 16.** Main technological players in the fields of $CO_2$ capture and purification for manufacture of chemicals.

| Company | Country | Technology | Technology Description | Scale | Patents Related to $CO_2$ Capture |
|---|---|---|---|---|---|
| Bright Energy | USA | Cryogenic carbon capture | Frost Carbon Capture (FrostCC) system that cools the flue gas to separate $CO_2$. | Bench | 0 |
| C-Capture | UK | Chemical absorption | Use of an amine-free solvent | Pilot | 2 |
| Carbon Clean Solutions | UK | Chemical absorption | Use of a proprietary solvent (APBS) that is able to capture flue gases with low $CO_2$ concentration; enabling over 90% of $CO_2$ capture from the feed flue gases. | Large | 8 |
| Carbon Engineering | | DAC (chemical absorption) | Use of a potassium hydroxide solution | Pilot | 9 |
| Carbozyme | USA | Enzymatically active membrane | Use of a polypropylene hollow fiber membrane with an immobilized catalyst (carbonic anhydrase, CA) | Bench | 0 |
| Climeworks | Swiss | DAC (adsorption) | Use of porous granulates modified with amines to capture of $CO_2$ from air. Desorption is performed by temperature swing. | Large (modular) scale | 9 |
| $CO_2$ Solutions | Canada | Enzymatically active solvent | Chemical absorption with immobilized enzymes (carbonic anhydrase, CA) | Large | 14 |
| FuelCell Energy | USA | MCFCs | MCFCs are electrochemical processes that are able to concentrate and capture $CO_2$ | Bench | 19 [a] |
| Global Thermostat | USA | Chemical adsorption | Use of amine-based chemical sorbents bonded to porous ceramic monoliths. $CO_2$ is stripped off and collected using low-temperature steam (85–100 °C). | Pilot | 5 [b] |
| InnoSepra | USA | Physical adsorption | Use of a zeolite-activated carbon by PSA and TSA. | Pilot | 2 |
| Innovator Energy | USA | Chemical absorption | Use of ammonia-based solution, which works at room temperature and pressure conditions. The organic solvent and $CO_2$ absorbing solution are then regenerated using low temperature heat. | Pilot | 0 |
| Jupiter Oxygen | USA | Oxyfuel combustion | Uses 95–100% pure $O_2$ as fuel to perform combustion. Highly concentrated $CO_2$ streams can be obtained afterward. | Large | 5 |
| Net Power | USA | Oxyfuel combustion | Using high-pressure oxyfuel, the supercritical $CO_2$ cycle is a $CO_2$ capture system based on the Allam cycle. | Pilot | 13 |
| Sunfire | Germany | SOFC | Cogeneration of heat and energy with easy separation of $CO_2$ | Large | 2 |
| Sustainable Energy Solutions | USA | Cryogenic carbon capture | $CO_2$ is cooled to about $-140$ °C), separated, pressurized, melted, and delivered at pipeline pressure | Small pilot | 12 |

[a] Patents directly related to $CO_2$ capture, although more than 125 patents are related to general aspects of the fuel cells. [b] Patents directly related to $CO_2$ capture; several others regarding DAC were found, but not counted. UK: United Kingdom; DAC: direct air capture; TSA: temperature swing adsorption.

The companies C-Capture, Carbon Clean Solutions, and Innovator Energy are developing new solvents for chemical absorption, especially amine-free solvents. Carbon

Clean Solutions is already operating at large scale. Particularly, in order to improve the $CO_2$ capture efficiency, this company is making use of carbonic anhydrase enzymes in the solvent. The enzyme-based system reduces the heat of absorption and reduces the energy penalty of the operation [118]. Following a similar idea, Carbozyme is supporting these enzymes on membranes for enhancement of separation efficiencies. Meanwhile, Carbon Engineering developed a chemical absorption process that uses a potassium hydroxide solution for $CO_2$ capture and that can be used for direct air capture (DAC).

DAC also constitutes the main subject of technological activities carried out by Climeworks and Global Thermostat, although using amine-based chemical sorbents. However, Global Thermostat, in partnership with ExxonMobil, is also using this technology to remove carbon dioxide from available industrial streams. In addition, InnoSepra is employing zeolite-activated carbon for $CO_2$ capture from flue gases. It has been reported, however, that adsorptive materials employed so far present fast adsorption kinetics, but low $CO_2$ adsorption capacity at low pressure. This certainly constitutes a significant limitation of the proposed process. Moreover, the high energy demand and long operation times required for $CO_2$ desorption currently reduce the competitiveness of the TSA approach [110].

Bright Energy and Sustainable Energy Solutions have proposed the use of cryogenic separation processes for capture and purification of $CO_2$, although the analyzed technologies have not yet been validated in large-scale industrial facilities, mainly because of the high energy demand of these processes [110]. In particular, Fuel Cell Energy (FCE) is the leading manufacturer of molten carbonate fuel cells, MCFCs, which has been applied to concentrate and capture $CO_2$ from combined flue gas and $CH_4$ streams (internally reformed to hydrogen), generating power and heat. According to this strategy, $CO_2$ can be captured from the depleted fuel stream that leaves the fuel cell [389,390]. Major drawbacks of this technology are the high maintenance and operation costs, although the high costs of MCFC units have been slowly reduced through the years, such that the technology can be regarded as promising [360].

Lastly, as already mentioned, oxycombustion processes can facilitate the $CO_2$ capture, although retrofitting of existing regular combustion sites can constitute a hard task. Jupiter Oxygen is developing oxyfuel combustion processes in partnership with Shell Cansolv. Moreover, 8 Rivers developed the Allam-Fetvedt cycle technology that is based on an oxyfuel combustion system and is somewhat similar to the technology developed by Net Power. These initiatives are promising to generate low-cost power from carbon-based fuels without producing air emissions [391]. Some partners of Net Power are Exelon, McDermott, Oxy Low Carbon Ventures, and Toshiba.

Patents Regarding $CO_2$ Capture Deposited by Plastic and Resin Manufacturers

A similar patent search was performed for regular and well-known plastic producers (as listed in Table 3), enabling the authors to state that most plastic producers are not very active in the technological development of $CO_2$ capture and separation processes (see Table 17). For instance, among the 20 large investigated companies, apparently half of them have not had the interest to deposit patents in this field: LyondellBasell, Lotte Chemical, LG Chem, PetroChina, LG Chem, Alpha Packaging, Borouge, Formosa Plastics, Sinopec, and Toray Industries. Moreover, Lanxess, INEOS, and Chevron Philips have not been very active in the field either, although we were able to detect one deposited patent for each of them. On the other hand, BASF is apparently the most active company in the field, with at least 24 deposited patents related to $CO_2$ capture and purification. As a matter of fact, BASF's good carbon management has been recognized by many organizations, with active participation in the "Cracker of the Future" consortium and in corporate climate actions [392]. (Six companies with petrochemical steam crackers in Europe have formed the "Cracker of the Future Consortium" to develop cracker technology with a carbon footprint. The companies seek to make the carbon savings by replacing natural gas–based boilers with ones that use renewable electricity. BASF, Borealis, BP, LyondellBasell Industries, Sabic and Total plan to have a pilot low-carbon-footprint cracker in operation by 2030 and

widespread commercial-scale production by 2050 [70].) ExxonMobil is also participating actively in the field, although we were unable to detect whether ExxonMobil's patents related to $CO_2$ capture and purification are indeed being used in the company's industrial crackers. Particularly, in 2019, ExxonMobil announced partnerships with FuelCell Energy, Global Thermostat, Synthetic Genomics, Mosaic Materials (young companies mentioned previously), and universities to reduce carbon emissions associated with energy generation [393–398].

Dow Chemical is developing amine-based chemical absorption processes for carbon capture from coal power plant-generated flue gas in partnership with Alstom Power, but does not seem directly involved with technologies intended to reduce carbon emissions from olefin manufacturing [399]. For instance, Dow Benelux (a subsidiary of Dow) is capturing $CO_2$ from a steel plant under the Carbon2Value project [400].

Lanxess deposited a patent on an amine-based solvent for $CO_2$ capture, although it has also deposited a patent for post-combustion $CO_2$ capture in partnership with Shell Cansolv. Additionally, the company announced recently the implementation of strategies to mitigate $CO_2$ emissions, mainly by using renewable energy sources [401].

Unlike most plastic producers, companies and organizations involved with refining operations, such as Shell, IFP, and Petrobras, have already developed and implemented robust $CO_2$ capture technologies. Several other companies have also developed technologies for $CO_2$ capture from flue gases, including Doosan, Alstom, Siemens, Thyssenkrupp, and Aker [402]. Another leader in $CO_2$ capture technologies is Linde, frequently in partnership with BASF. Particularly, Linde and SABIC built and operate a liquefaction plant to capture $CO_2$ from ethylene glycol production [59]. Moreover, Membrane Technology and Research possesses a wide portfolio of membranes that are suitable for $CO_2$ capture [114], although these technologies are not yet employed on a large commercial scale. Air Liquide and Air Products are also proprietary of membrane technologies, including an interesting hybrid membrane–cryogenic distillation process by Air Liquide [331,350,403]. Additionally, Mitsubishi Heavy Industries operates $CO_2$ capture plants that are coupled to coal-fired power stations and natural gas-fired steam reformers, although none of the plants are related to olefin manufacturing [404,405].

As a whole, although many companies have already developed efficient $CO_2$ capture technologies, the implementation of these technologies in industrial steam cracking facilities or other industrial sites related to monomer synthesis is indeed very scarce. This is probably due to the fact that energy generation sites and processes produce significantly larger amounts of $CO_2$ in exhaust streams, constituting the main locus of technological development in the field at the moment. Nevertheless, this shows that a lot of work is ahead of major plastic producers to reduce $CO_2$ emissions in industrial sites.

**Table 17.** Technology description of $CO_2$ capture solutions developed by chemical and plastic producers.

| Company | Country | Technology | Technology Description | Scale | Patents in $CO_2$ Capture |
|---|---|---|---|---|---|
| BASF | Germany | Chemical absorption | Chemical absorption mainly with amines (especially methyldiethanolamine), but also ammonia. | Large scale | 24 |
| ExxonMobil | USA | Adsorption; MCFCs | Amine or zeolite-based materials for adsorption of $CO_2$. Desorption may be performed by a swing adsorption process. Development of MCFCs is also occurring in partnership with FuelCell Energy, Inc. | ℀ | 23 |
| Sumitomo Chemical | Japan | Membranes | Acid gas membrane separation to remove $CO_2$ from synthesis gas, natural gas, exhaust gas, and gas streams that contain $N_2$ and $O_2$. Membranes contain at least one hydrophilic polymer layer. | Pilot unit | 9 |

**Table 17.** *Cont.*

| Company | Country | Technology | Technology Description | Scale | Patents in $CO_2$ Capture |
|---|---|---|---|---|---|
| Reliance Industries | India | Absorption; adsorption | Chemical absorption with ionic liquid. Adsorption with oxides (temperature swing). | ✿ | 8 |
| ENI | Italy | Algae | Cultivation of microalgae for posterior biomass bioconversion into ethanol, butanol, and diesel. | Pilot | 5 |
| Dow Chemical | USA | Adsorption | Utilization of amine/alkanolamine for $CO_2$ removal. | Pilot unit | 3 |
| Mitsubishi Chemical ▲ | Japan | Membrane | Zeolite membrane for removal from methane. | ✿ | 3 |
| Sabic | Saudi Arabia | Cryogenic separation; Chemical absorption | Removal of $CO_2$ from syngas using aqueous solvents or cryogenic separation. | ✿ | 2 |
| Chevron Phillips | USA | Physical absorption | Use of ionic liquids to separate $CO_2$. | ✿ | 1 [406] |
| Ineos | UK | Chemical absorption | Chemical absorption with amines. | Bench | 1 |
| Lanxess | Germany | Chemical absorption | Absorption using polystyrene-based resins that contain primary amines and are crosslinked with divinyl aromatics. | Pilot unit | 1 |

✿ TRL was not found. ▲ Patents of Mitsubishi Heavy Industries and Dow Global Technologies Llc were not included in the search.

### 3.4. Bibliometric Analysis of Carbon Utilization: Technical Information

In order to attend the policies to control and reduce GHG emissions, the captured carbon dioxide must be used somehow. Transportation and subsequent underground storage of $CO_2$ normally constitute the main strategy, being particularly useful for enhanced oil recovery (EOR) by oil and gas industries. However, the true capacity to store $CO_2$ safely has become subject of heated debates. Furthermore, the continuous and rising rates of gas and oil extraction are not sustainable even if increasing amounts of $CO_2$ are captured and stored, if one considers the cradle-to-grave lifecycles [29,161]. Therefore, alternative strategies that can contribute to a reduction in the rates of gas and oil extraction are certainly welcome. For this reason, development of new uses for captured $CO_2$ is advisable. Currently, the commercial utilization of $CO_2$ focuses basically on the manufacture of urea, salicylic acid, polycarbonates, and polyurethanes, as alternatives to the use of $CO_2$ as supercritical fluid [92]. However, many more attractive opportunities are available.

As a matter of fact, $CO_2$ is an attractive building block, as it can be converted into valuable chemicals that can find many interesting uses [29,161]. Synthetic fuels ("e-fuels"), chemical feedstocks, and polymers are some typical examples, as shown in Figure 20 [407]. Among these many possible products, most of them are already in use, rendering the decarbonization of the economy by $CO_2$ recycling possible and feasible. Consequently, when $CO_2$ is treated as a valuable raw material that must be converted into a valuable chemical product, this strategy is certainly in agreement with the principles and expectations of the circular economy transition.

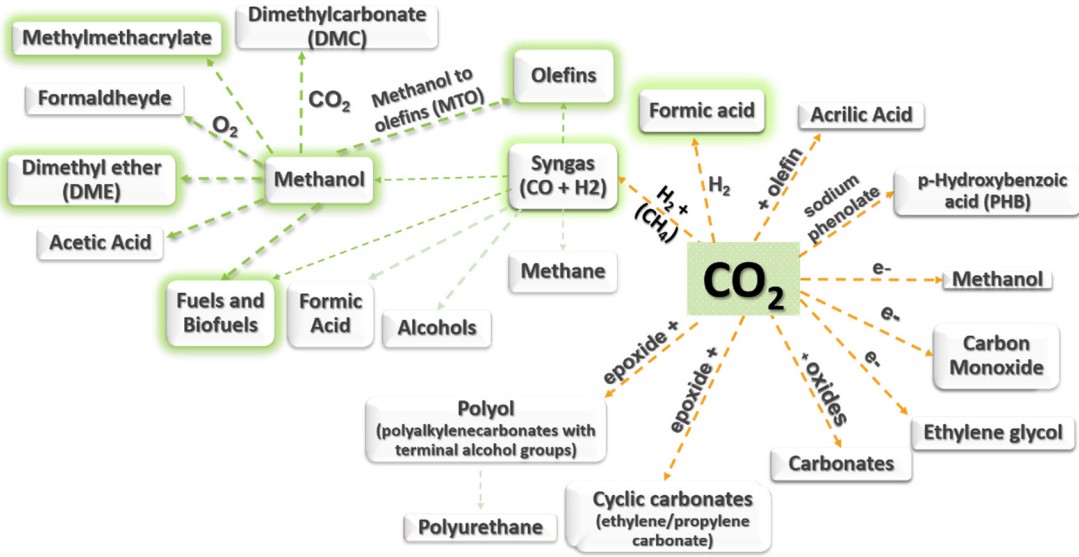

**Figure 20.** Products that can be currently manufactured using $CO_2$ as a building block.

CO$_2$ utilization (CDU) constitutes a significant and well-known challenge, due to high stability of the molecule and, consequently, the high energetic demand of most $CO_2$ transformation processes. Despite that, the thermodynamic and kinetic barriers are being surpassed with the help of better catalysts and processes [29]. Overall, most CDU processes make use of carboxylation and reduction reactions using different and well-established thermochemical, electrochemical, or photochemical reactions. Carboxylation takes place when the $CO_2$ transformation reaction does not involve the complete breakage of the carbonyl double bonds (C=O), as observed in mineral carbonation [408] and some polymerization reactions [409]. However, mineral carbonation does not lead to interesting applications in the plastics industry, although it can be used to reduce $CO_2$ emissions through solid mineral storage of carbon. Therefore, even though mineral carbonation has been intensively investigated to mitigate $CO_2$ emissions, further discussions are not provided in the present work.

On the other hand, during the reduction reaction (through regular catalytic, biocatalytic, photocatalytic, and electrocatalytic processes) at least one carbonyl (C=O) double bond is broken. Due to the high stability of $CO_2$, the reaction is energetically demanding (requiring high amounts of energy provided by sunlight, heat, electricity, or microwaves), and it requires the use of sophisticated catalysts (thermocatalysts, biocatalysts, photocatalysts, electrocatalysts, or combinations of these catalysts) and highly energetic reactants (such as H$_2$) [122]. Nevertheless, the $CO_2$ reduction process can generate methane, synthetic gas (syngas, H$_2$ and CO), methanol, dimethyl ether (DME), ethanol, and formic acid, among other valuable chemical platforms, enabling the insertion of carbon into the productive chemical cycle, managed mainly by chemical industries and used to produce plastics. Particularly, syngas has long been used as an intermediate for the manufacture of several hydrocarbon products (such as olefins) through the well-known and mature Fischer–Tropsch (FT) transformation route, as shown schematically in Figure 21 [158]. This gives technological experience, maturity, and robustness for $CO_2$ transformations into other valuable chemical products.

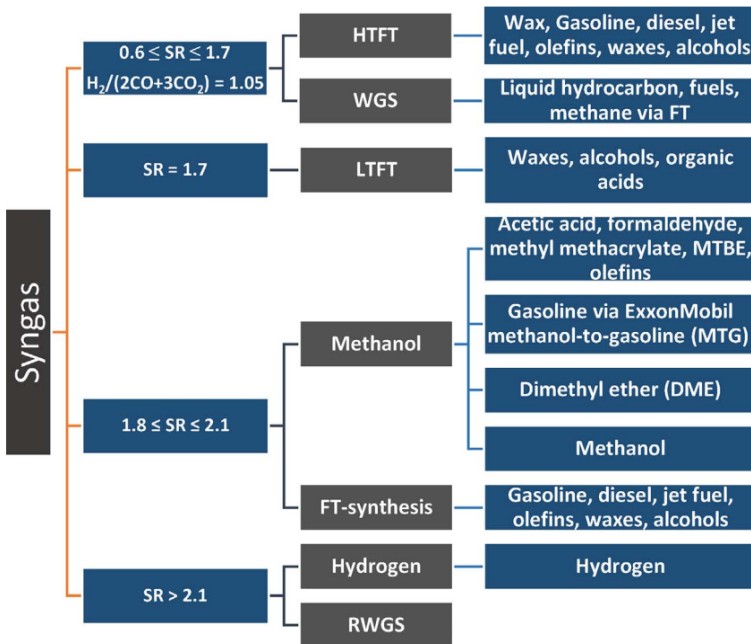

**Figure 21.** Products that can be obtained from syngas. SR: steam reforming; HTFT: high-temperature Fischer–Tropsch synthesis; LTFT: low-temperature Fischer–Tropsch synthesis; WGS: water gas shift reaction; RWGS: reverse water gas shift; FT: Fischer–Tropsch. Reproduced with permission from Priyadarshini Balasubramanian, Ishan Bajaj, M.M. Faruque Hasan, Journal of $CO_2$ Utilization; published by Elsevier, 2018 [410].

　　Bioconversion of biomass produced by bacteria, enzymatic processes, and microalgae has been thoroughly investigated. However, $CO_2$ bioconversion has been discussed previously, where it was shown that the technique still needs much development to become economically feasible and is difficult to retrofit. For these reasons, bioconversion is expected to become competitive in the medium or long term [92]. Nonetheless, bioconversion can be indeed very advantageous and competitive in some cases when it involves the production of ethanol, which also constitutes a well-established and mature raw material for manufacture of chemicals in general and monomers in particular, including ethylene and succinic acid [136,388]. Regarding more specifically the production of olefins, steam cracking and direct dehydrogenation of molecules of higher molar masses remain the technologies normally used at present. However, both technologies require the use of high temperatures and depend on the availability of fossil fuels. For these reasons, it can be expected that new trends and developments and availability of cheaper chemical bioplatforms (such as ethanol) will change this scenario in the near future. Figure 22 presents some $CO_2$ chemical conversion pathways that can be considered interesting for plastic manufacturers. One must consider that technologies that allow the production of syngas or methanol as intermediates or allow the direct manufacture of olefins can be regarded as the most valuable ones by current standards.

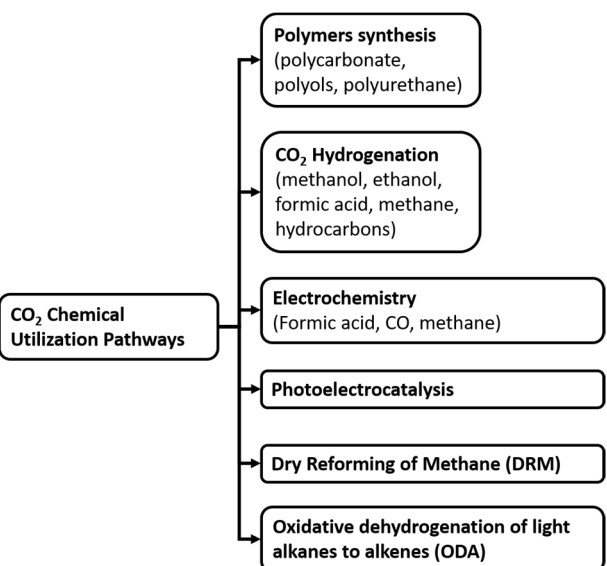

**Figure 22.** Main $CO_2$ transformation processes and resulting products. Adapted from: [156].

Table 18 summarizes the most cited CDU processes, as observed in the scientific literature. Although enhanced oil recovery (EOR), enhanced coal bed methane recovery (ECBM), mineralization, and bioconversion (by cyanobacteria, microalgae, gas fermentations with anaerobic acetogens, microbial electrosynthesis, and hybrid systems) account for 39.5% of the citations, they were not considered appropriate for carbon recycling in plastics industries in the present text. In addition to the specific processes listed in Table 18, other chemical reactions that appeared more frequently include urea synthesis, direct carboxylation of olefins, carboxylation of hydrocarbons (methanol, methane, or benzene), reactions with oxetane, isocyanate, and carbamate, synthesis of oxazolidinone and quinazoline, and the reaction with ethylene to produce ethylene oxide [126,158,293,411]. Direct use of $CO_2$ also appeared in 11.4% of the citations and includes the use of supercritical $CO_2$ and applications in the food and beverage industries. Regarding the enzymatic conversion of $CO_2$, many technological barriers must still be surpassed before the development of commercial applications, although enzymes are already being applied for $CO_2$ capture [332]. In addition, the use of carbon dioxide instead of $O_2$ to perform the oxidative coupling of methane (OCM) and manufacture ethylene has also been suggested and investigated, although low $CH_4$ conversions have been reported so far [177,412].

Figure 23 illustrates the frequency distribution of citations of the different carbon dioxide utilization processes through the years. Hydrogenation was the second most cited CDU transformation process in the period, but the number of citations fluctuated significantly through the years due to the development of better catalysts, improved processes, and cheaper and greener hydrogen sources. A similar behavior was observed with other technologies, including the direct polymerization or cycloaddition of $CO_2$ to produce cyclic carbonates, polyethercarbonates, polycarbonates, and polyurethanes. These polymerization processes are developing fast and count on already well-developed markets to become a reality. Meanwhile, a fast increase in citations of electrochemical and bioconversion processes is evident after 2010, possibly indicating the most important trends for the near future. Nevertheless, it is important to note that the existence of commercial large-scale electrochemical facilities could not be detected and that some technological barriers limit the growth of bioconversion processes at the present moment, as discussed before. The technologies shown in Table 18 and Figure 23 are explained in the sections below.

**Table 18.** Carbon dioxide utilization processes cited most frequently.

| Carbon Utilization | No. of Identified Processes | Percentage (%) | References |
|---|---|---|---|
| Bioconversion | 66 | 16.7 | [95,126,137,142,145,146,197,253–301] |
| Hydrogenation | 51 | 12.9 | [92,95,108,114,117–119,142,158,176,177,180,182,194,208,209,233,235,266,284,285,293,330,340,345,413–425] |
| Mineralization | 48 | 12.2 | [95,105,116,117,126,142,150,151,154,157,158,180,187,273,284,285,293,298,302–324] |
| Other reactions with $CO_2$ | 45 | 11.4 | [95,105,117,126,158,176,183,201,208,225,235,266,273,284,285,293,345,426–430] |
| EOR and/or ECBM | 42 | 10.6 | [95,104,105,126,142,148,158,174,182,198,214,222,227,235,273,284,293,302,303,416,431–449] |
| Electrochemical reduction | 34 | 8.6 | [95,108,109,142,158,186,262,266,273,289,293,300,330,336,340,415,416,418,423,450–459] |
| Polymerization or cycloaddition | 25 | 6.3 | [73,158,176,177,180,197,201,208,234,235,266,273,284,285,293,415,460–466] |
| Direct use | 23 | 5.8 | [73,95,104,108,144,158,177,182,201,273,285,293,299,467] |
| Dry reforming of methane | 17 | 4.3 | [73,92,108,117,118,177,180,194,284,293,416,423,424,433,461,468] |
| Methanation | 14 | 3.5 | [110,117,118,139,142,266,293,424,439,469–471] |
| Photocatalytic ERC | 8 | 2.0 | [118,147,149,415,469,472–474] |
| Tri-reforming of methane | 8 | 2.0 | [73,92,142,177,266,293,415,469] |
| Enzymatic conversion | 6 | 1.5 | [272,299,330,332,475,476] |
| Oxidative dehydrogenation of alkanes (ODH) | 4 | 1.0 | [177,180,293,461] |
| Carnol | 2 | 0.5 | [108,177] |
| OCM | 2 | 0.5 | [177,461] |

OCM: oxidative coupling of methane.

### 3.4.1. Electrochemical Cells

Chemical synthesis through the $CO_2$ electroreduction reaction ($CO_2$-RR, or electrochemical reduction of $CO_2$, ERC) constitutes a promising technology for manufacture of many chemical compounds, including synthetic fuels ("electro-fuel" or a "carbon-based electro fuel") and oxygenates [452]. The RR reaction between $CO_2$ and water can be conducted in an electrolyzer at ambient conditions, controlling the reaction rate through manipulation of the overpotential. According to the number of electrons transferred per molecule of $CO_2$ during the reaction, the electrochemical reduction of $CO_2$ can give birth to many distinct electron pathways in aqueous and nonaqueous electrolytes. Some of the major obtained products and respective thermodynamic electrochemical half-reactions are shown in Table 19. Some of the main products are oxalic acid, CO, formic acid, formaldehyde, methane, methanol, ethane, ethylene, and ethanol. However, several side reactions can also take place, leading to reduced faradaic and energy efficiencies [158]. Consequently, the search for more efficient electrocatalysts, or combinations of photo- and electrocatalysts, has been intense and several attempts are currently being performed to increase the yields

of hydrocarbons and light olefins [477]. Many reviews about ERC processes are available elsewhere [119,158,357,407,452–454,478,479].

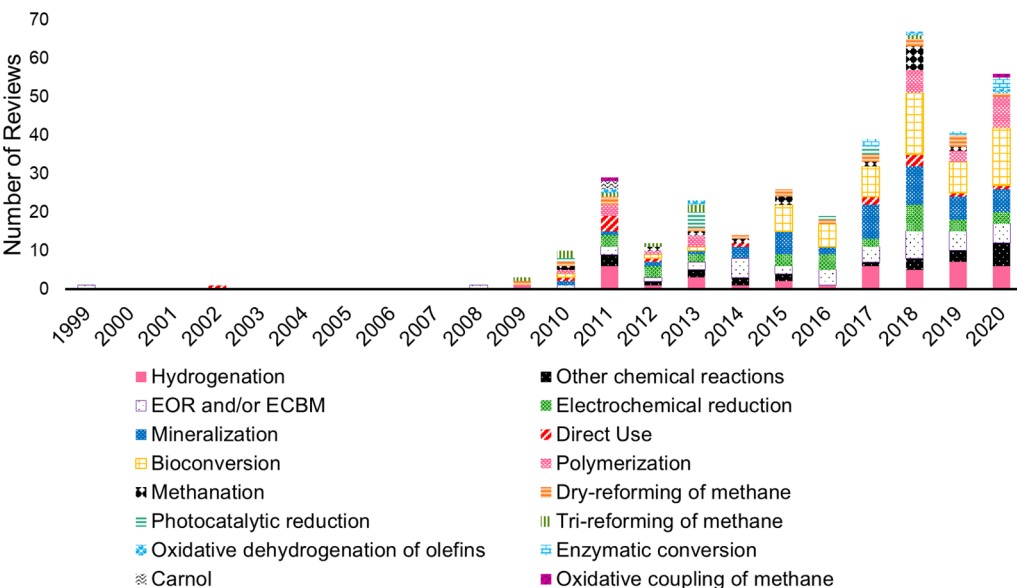

**Figure 23.** Frequency distribution of citations of carbon dioxide transformation processes through the years.

One of the advantages of $CO_2$-RR is the much easier post-reaction separation of the main products, as they are produced at different electrodes [407,452]. Nonetheless, it must be emphasized that electrochemical processes are energetically demanding, such that these processes are sustainable only when renewable or non-fossil energy can be used. De Luna et al. (2019) discussed some important points that must be considered to perform the transition to neutral carbon-emitting chemical production through renewably powered electrosynthesis. Although some scientific and engineering challenges must still be solved in this area, the authors claimed that "*the economics of electrocatalytic processes will be highly dependent on the availability and price of renewable electricity, the regional cost of feedstock and of traditional petrochemical manufacture, the maturity of carbon capture technologies, and the social, political, and economic incentives to transition to low-carbon processes*" [480].

Valderrama et al. (2019) [35] proposed the use of electrochemical cells for manufacture of formic acid and posterior manufacture of polyesters of oxalic and glycolic acid. However, significant research is still required to optimize the process and deal with the challenges related to manufacturing costs, low product selectivities, and high energy demand [35].

Pappijin et al. (2020) [481] studied the viability of the direct electrochemical conversion of $CO_2$ to ethylene and concluded that, if combined with green electricity, "*the electrochemical reduction of $CO_2$ can lead to a negative overall $CO_2$ balance, but several breakthroughs are needed to make this competitive with the current state of the art under current market conditions*", requiring a decrease of the electricity price and larger capacities in renewable electricity production [481]. Moreover, improvements in catalyst performance and a reduction in installation and maintenance costs associated with the electrochemical process are also required [481].

Contrary to electrocatalysis, photochemical processes (photochemical or photocatalytic) can be very selective toward a product. In spite of that, many technological challenges still prevent the widespread use of photo-assisted $CO_2$ reduction processes, as discussed elsewhere [158,469,482].

**Table 19.** Half-cell reaction pathways for carbon dioxide electroreduction and respective thermodynamic potentials (V vs. SHE; i.e., electrode potentials vs. "standard hydrogen electrode") for reactions performed in aqueous solution, at 298 K and 1.0 atm. Reproduced with permission from Jinli Qiao, Yuyu Liu, Feng Hong, Jiujun Zhang, Chemical Society Reviews; published by Royal Society of Chemistry, 2014 [483].

| Thermodynamic Electrochemical Half-Reactions | Electrode Potentials (V VS. SHE) |
|:---:|:---:|
| $CO_2(g) + 2H^+ + 2e^- \rightarrow CO + H_2O\ (l)$ | $-0.106$ |
| $CO_2(g) + 6H^+ + 6e^- \rightarrow CH_3OH\ (l) + H_2O\ (l)$ | $0.016$ |
| $CO_2(g) + 5H_2O(l) + 6e^- \rightarrow CH_3OH\ (l) + 6\ OH^-$ | $-0.812$ |
| $CO_2(g) + 8\ H^+ + 8e^- \rightarrow CH_4\ (g) + 2H_2O\ (l)$ | $0.169$ |
| $CO_2(g) + 6\ H_2O(l) + 8e^- \rightarrow CH_4\ (g) + 8\ OH^-$ | $-0.659$ |
| $2\ CO_2(g) + 12\ H^+ + 12\ e^- \rightarrow CH_2CH_2\ (g) + 4H_2O\ (l)$ | $0.064$ |
| $2\ CO_2(g) + 8\ H_2O(l) + 12\ e^- \rightarrow CH_2CH_2\ (g) + 8\ OH^-$ | $-0.764$ |
| $2\ CO_2(g) + 12\ H^+ + 12\ e^- \rightarrow CH_3CH_2OH\ (l) + 3H_2O\ (l)$ | $0.084$ |
| $2\ CO_2(g) + 9\ H_2O(l) + 12\ e^- \rightarrow CH_3CH_2OH\ (l) + 12\ OH^-$ | $-0.744$ |

### 3.4.2. Dry Reforming of Methane (DRM)

The steam reforming of methane (SMR) and partial oxidation reforming (POR, POx or POM) are the methods used most often for the production of syngas (see Table 20). Despite that, other less energy-demanding processes (including autothermal reforming, ATR, a combination of SMR and POx) can be used for manufacture of syngas, such as the dry reforming of methane (DRM), which is based on the reaction of $CO_2$ with $CH_4$ [484]. DRM advantages include the higher syngas purity (when compared to other reforming technologies, due to very high methane conversion, frequently higher than 98%) and the lower operating costs (around 20%) [117,194]. Nevertheless, development of more active and more stable catalysts is still required due to problems usually associated with the high operating temperatures, such as the formation of coke [114,411]. Particularly, the combined tri-reforming of methane, a combination of SMR, POR, and DRM, can provide syngas with controllable $H_2/CO$ ratios (1.5–2.0), which can be very important for FT-based processes. Additional benefits also include the lower energy requirement and the lower rates of coke formation due to the presence oxygen [415,469]. As a whole, it can be said that syngas production through a reaction with methane probably constitutes the simplest alternative to use $CO_2$ for manufacture of other chemicals at the present time [108,411]. Moreover, biogas (in essence a mixture of methane and carbon dioxide) is widely available in landfills and anaerobic digestion processes [411].

**Table 20.** Main reactions involved in the catalytic reduction of methane with $CO_2$.

| Reaction. | Stoichiometry | Enthalpy |
|:---:|:---:|:---:|
| Steam methane reforming (SMR) | $CH_4 + H_2O \leftrightarrow CO + 3H_2$ | $\Delta H^\circ_{298\ K} = 206.2\ \text{kJ/mol}$ |
| Partial oxidation reforming (POR), partial oxidation of methane (POM), or partial oxidation (POx) of methane | $CH_4 + \frac{1}{2}O_2 \leftrightarrow CO + 2H_2$ | $\Delta H^\circ_{298\ K} = -35.6\ \text{kJ/mol}$ |
| Dry methane reforming (DRM) | $CH_4 + CO_2 \leftrightarrow 2CO + 2H_2$ | $\Delta H^\circ_{298\ K} = 247.3\ \text{kJ/mol}$ |

Moreover, methane can be used to produce methanol through the Carnol process (see Table 21), which initially pyrolyzes methane to produce $H_2$ and solid carbon (>800 °C) [108]. Then, the hydrogen stream reacts with $CO_2$ to yield methanol and other hydrocarbons (as described in the sections below). It must be emphasized that the methane decomposition can also be performed simultaneously with the DRM process [108]. Furthermore, it must be observed that applications of the solid carbon byproduct have not been fully developed yet.

**Table 21.** Main reactions involved in the Carnol process.

| Reaction | Stoichiometry | Enthalpy |
|---|---|---|
| Methane thermal decomposition | $CH_4 \leftrightarrow C + 2H_2$ | $\Delta H^\circ{}_{298\ K} = 17.9$ kcal/mol |
| *Carnol* process | $3CH_4 + 2CO_2 \leftrightarrow 2CH_3OH + 2H_2O + 3C$ | - |
| Methane thermal decomposition + DRM | $2CH_4 + CO_2 \leftrightarrow 2CO + 4H_2 + C$ | - |

- Information was not found.

Lastly, as mentioned previously, carbon dioxide oxidative coupling of methane ($CO_2$ OCM) constitutes another evolving field of research, enabling the production of ethylene directly through the catalytic reaction of methane (see Table 22). However, this reaction is only feasible at a very high temperatures, while it still depends on an improvement of catalysts and optimization of operation conditions to become economically viable [177,412,485,486].

**Table 22.** Main reactions involved in the OCM and $CO_2$ OCM processes.

| Reaction | Stoichiometry | Enthalpy |
|---|---|---|
| *OCM* | $2\ CH_4 + O_2 \leftrightarrow C_2H_4 + 2\ H_2O$ | $\Delta H^\circ{}_{298\ K} = -280$ kJ/mol |
| *CO_2 OCM* | $2\ CH_4 + 2\ CO_2 \leftrightarrow C_2H_4 + 2\ CO + 2\ H_2O$ | $\Delta H^\circ{}_{298\ K} = -284$ kJ/mol |
| | $2\ CH_4 + CO_2 \leftrightarrow C_2H_6 + CO + H_2O$ | $\Delta H^\circ{}_{298\ K} = +106$ kJ/mol |

### 3.4.3. Hydrogenation

Hydrogenation processes can be used to manufacture oxygenates and/or hydrocarbons and are related to the classic reverse water gas shift reactions (rWGS), which makes $CO_2$ conversion through this route more feasible, as this is probably the most well studied area for $CO_2$ conversion. In fact, rWGS is the most important route for $CO_2$ utilization, because CO is a raw material widely used for production of methanol and other hydrocarbons. The main bottlenecks of this process are the endothermic nature of the reaction, the usual low conversions attained at moderate temperatures, and the relatively low product selectivities, although the development of new catalysts is quickly changing this characteristic feature of hydrogenation processes. Liu et al. [487] recently reviewed the fields of heterogeneous catalysis and plasma catalysis associated with carbon hydrogenation and production of valuable chemicals.

The production of oxygenates (especially alcohols and ethers, such as methanol and DME) through hydrogenation processes constitutes an advanced and mature technological field, reporting the operation of demonstrative, pilot-scale, and commercial-scale plants [488,489]. This opens a bright future for the methanol economy, as described by Olah [108]. According to Olah, methanol should be transformed into a chemical platform for manufacture of many other products, including, for example, formaldehyde, DME, acetic acid, methylamines, methyl *tert*-butyl ether, methyl methacrylate, polyalcohols, and silicones. Furthermore, methanol can also be used as fuel for energy generation. According to this scenario (see Table 23), methanol can be produced from $CO_2$ both directly (through direct $CO_2$ hydrogenation) and indirectly (by converting syngas into methanol, when rWGS is followed by CO hydrogenation, for instance).

**Table 23.** Main reactions involved in the production of methanol from $CO_2$.

| Reaction | Stoichiometry | Enthalpy |
|---|---|---|
| CO hydrogenation | $CO + 2H_2 \leftrightarrow CH_3OH$ | $\Delta H^\circ{}_{298\ K} = -90.6$ kJ/mol |
| Methanol synthesis | $CO_2 + 3H_2 \leftrightarrow CH_3OH + H_2O$ | $\Delta H^\circ{}_{298\ K} = -49.5$ kJ/mol |
| rWGS | $CO_2 + H_2 \leftrightarrow CO + H_2O$ | $\Delta H^\circ{}_{298\ K} = 41.2$ kJ/mol |

According to Álvarez et al. (2017) [119], the total capital investment needed to build a methanol plant that makes use of $CO_2$ and $H_2$ as raw materials is estimated to be similar to the total capital investment needed to build a conventional syngas-based plant, as both plants are based on packed-bed reactors [119]. Demonstration plants for the direct synthesis of methanol by hydrogenation of $CO_2$ and $H_2$ are in operation in Iceland and Japan [34,490]. The main problems related to the direct manufacture of methanol are activation of the C–H bond with currently available Cu-based catalysts and the little availability of cheap sources of hydrogen and $CO_2$ [114,119].

On the other hand, production of hydrocarbons through well-known FT processes can be limited by the existing thermodynamic barriers and the higher energetic demands. Consequently, the main issues that affect these processes are related to the lower conversion efficiencies and energy costs [491]. Furthermore, although $H_2$ is necessary for the production of oxygenates, manufacture of hydrocarbons demands much more hydrogen per unit of product than the formation of oxygenates (see Table 24) [92,491]. Therefore, $H_2$ required for hydrogenation must be produced with renewable resources (such as methane from renewable sources) in order to be profitable and sustainable [2,34,491,492]. Some renewable hydrogen sources that can be considered are as follows:

i.　　Production with electrochemical cells (water splitting) using renewable energy (power-to-gas technology, P2G) [493,494];

ii.　　Methane decomposition into hydrogen and solid carbon through thermal pyrolysis, avoiding the production of $CO_2$ as a byproduct (although applications of the solid carbon byproduct are not fully developed and heat transfer problems must still be solved in the moving carbon bed reactor) [68];

iii.　　Biomass thermochemical decomposition in presence (gasification) or absence (pyrolysis) of oxygen at elevated temperatures, although the technological development of the field is still immature in many aspects and the process operation and blends of obtained products are highly dependent on the quality of available feedstocks [495], justifying the chemical looping gasification as an alternative strategy [430].

**Table 24.** Main reactions and products involved in $CO_2$ hydrogenation processes.

| Reaction | Stoichiometry |
|---|---|
| Waxes | $nCO + (2n+1)H_2 \leftrightarrow C_nH_{2n+2} + nH_2O$ |
| Olefins | $nCO + (2n)H_2 \leftrightarrow C_nH_{2n} + nH_2O$ |
| Methane | $CO + 3H_2 \leftrightarrow CH_4 + H_2O$ |
| Alcohols | $nCO + (2n)H_2 \leftrightarrow C_nH_{2n+1}OH + (n-1)H_2O$ |
| Carbon deposition | $2CO \leftrightarrow C_{(s)} + CO_2$ |

FT reactions are used to produce a wide spectrum of hydrocarbons that can be further refined to gasoline, jet fuel, and diesel (as in the case of the syngas-to-methanol process, SGTM). FT reactions can also be used to synthetize olefins, although better multifunctional catalysts are still required to avoid the occurrence of side reactions and increase the product selectivities [496–498]. For this reason, the use of multiple reactors and multiple chemical transformation steps (usually though initial production of methanol) has been frequently proposed and investigated [92,499]. Some important chemical platforms can be produced through these processes, including acetic acid, formaldehyde, and aromatics. In these cases, the $H_2$:CO ratios, reaction temperatures, reaction pressures, and employed catalysts significantly affect the relative yields of the product streams. Preferably, the syngas used as raw material for the methanol or Fischer–Tropsch syntheses should present $H_2$:CO molar ratios between 1.5:1 to 2.6:1 [410,500]. Additionally, the reaction should be preferentially carried out at pressures between 15 and 50 bar [500] and temperatures between 200 and 300 °C [501].

Overall, supported Ni-, Ru-, Rh-, Cu-, or Co-based catalysts on different support oxides ($TiO_2$, $SiO_2$, $Al_2O_3$, $CeO_2$, $ZrO_2$) have been used to perform $CO_2$ hydrogena-

tion reactions, enabling the manufacture of a wide range of products [118,119,470,502]. Methanol, for example, is currently produced using copper–zinc oxide/chromium oxide catalysts [503], while methane is the main product during the reaction over Ni and Ru catalysts [502]. Thus, as one can see in Figure 24, hydrogenation constitutes a promising route for manufacture of valuable chemicals from $CO_2$ and methanol, as described previously and in the sections below.

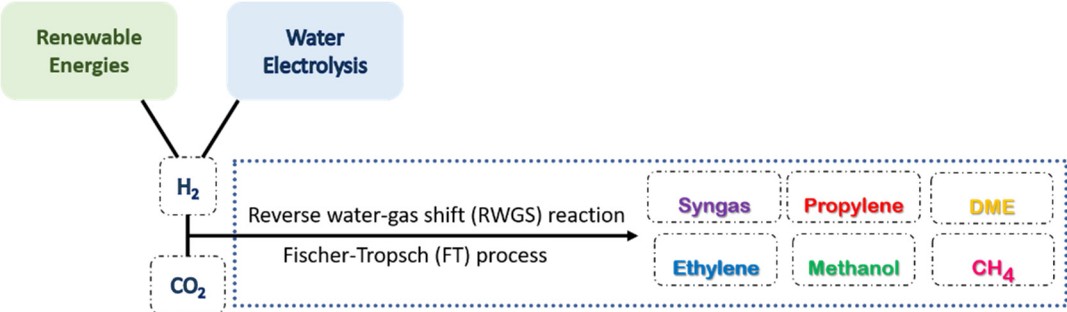

**Figure 24.** Main products from $CO_2$ hydrogenation.

### 3.4.4. Methanation

Methanation, or the Sabatier reaction, is the catalytic hydrogenation of $CO_2$ or CO to methane, as shown in Table 25 [471]. The reaction is favored at high pressures (10–30 atm) and at low temperatures (lower than 500 °C). Nanosized nickel particles (such as $Ni/Al_2O_3$) are those used most often to perform these reactions and are the commercially available catalysts normally employed for methanation (supplied by manufactures such as Johnson Matthey, Topsøe, and Clariant-Süd Chemie) [504,505]. However, the use of several other catalysts has also been investigated. For instance, Navarro et al. (2018) described a catalyst ordered sequence of decreasing activities in the form Ru > Ni > Co > Fe > Mo for the active phase, and $CeO_2$ > $Al_2O_3$ > $TiO_2$ > MgO for the active support [376].

**Table 25.** Main reactions involved in the production of methane through $CO_2$ hydrogenation.

| Reaction | Stoichiometry | Enthalpy |
| --- | --- | --- |
| Sabatier reaction | $CO_2 + 4H_2 \leftrightarrow CH_4 + 2H_2O$ | $\Delta H^\circ_{298\ K} = -164.7$ kJ/mol |
| FT synthesis | $CO + 3H_2 \leftrightarrow CH_4 + H_2O$ | $\Delta H^\circ_{298\ K} = -206$ kJ/mol |

At the present moment, only pilot plants are in operation for recovery of $CO_2$ through methanation strategies, such as those described by Audi and Sunfire [470], due to the still low conversions and product yields and necessity to improve the reaction heat management [471,505]. The economic viability is also very dependent on the hydrogen costs [470]. Despite these points, $CO_2$ hydrogenation to $CH_4$ through catalytic reactions constitutes a promising technological route, as discussed elsewhere [470,471,493,494].

Palm et al. (2016) studied the viability of "electricity-based plastics" considering the Sabatier reaction and using hydrogen obtained from water through solid oxide electrolysis cells (SOECs) [25]. The methane should be used afterward for production of ethylene through oxidative coupling of methane (OCM), while propylene can be produced through the methanol-to-propylene (MTP) process, as described below. Using these routes, roughly 3 tons $CO_2$/ton ethylene or propylene would be needed. The authors concluded that a complete shift to electricity-based plastics is possible from a technological point of view, but that production costs must be reduced 2–3-fold for the process to be economically viable, as the technology depends significantly on energy costs (70% of the costs are related to consumption of electricity).

### 3.4.5. Methanol-To-Olefin (MTO) Reaction

As discussed previously, at the present moment, olefins are mostly produced through cracking of naphtha or NGL reactions. However, while the combination of biofixation and catalytic cracking of biomass will certainly constitute an important technological alternative for manufacture of olefins in the near future, methanol-to-olefin (MTO), oxygenate to olefin (OTO), and methanol-to-propylene (MTP) reactions also constitute alternative routes for manufacture of light olefins ($C_2$ and $C_3$) [506–508].

MTO mechanisms are not yet fully clear, although important advances have been achieved in this field [509,510]. For instance, dimethyl ether (DME) can be produced through the dehydration of methanol over alumina ($Al_2O_3$) and posteriorly reacted with methanol with the help of zeolite ZSM-5 or silico aluminophosphate (SAPO-34) catalysts, yielding olefins ranging from $C_2$ to $C_8$ [158]. Current research activities are concentrated on development of new catalysts that can provide higher conversions, higher selectivities to ethylene and propylene, and long-term stability, among other aspects [158,503,506,511]. A review on recent developments of MTO and MTP processes was recently published by Yang et al. (2019) [510]. MTO processes have been performed on large commercial scales for at least 10 years by the Dalian Institute of Chemical Physics (DICP), Norsk Hydro, UOP, and Lurgi [388,510,512,513]. Specifically for capture and use of $CO_2$ emission, a few coal- or natural-gas-based operating plants can be found in China [514,515].

### 3.4.6. Polymerization (Polycarbonates and Polyurethanes)

Production of cyclic carbonates through reactions between $CO_2$ and epoxides (often ethylene oxide, propylene oxide, and cyclohexene oxide) [158], shown in Figure 25, is well established and dates back to the 1950s, but its commercial importance is currently increasing. Particularly, new catalysts are being developed for production of cyclic carbonates in order to decrease the pressure and temperature requirements and increase the selectivity toward the desired products [516]. Additionally, ethylene oxide can be formed through the reaction between ethylene and $CO_2$, forming CO as a byproduct, which is a valuable raw material for manufacture of methanol and for other FT reactions, as discussed previously [517]. Aromatic polycarbonates can be posteriorly manufactured through polymerization of cyclic carbonates [411,415], which constitutes an advantageous alternative to the oxidative carboxylation route that involves reactions of olefins, $CO_2$, and oxygen [503].

$$CO_2 + \text{(epoxide)}-R \longrightarrow \text{(cyclic carbonate)}-R$$

R = H      for ethylene carbonate

R = CH$_3$  for propylene carbonate

**Figure 25.** Synthesis of ethylene carbonate through reaction between $CO_2$ and ethylene oxide. Reproduced with permission from Martina Peters, Burkhard Kohler, Wilhelm Kuckshinrichs, Walter Leitner, Peter Markewitz, Thomas E. Muller, ChemSusChem; published by John Wiley and Sons, 2011 [158].

Polyurethanes can be produced through the reaction of $CO_2$ with cyclic amines (usually aziridines and azetidines) [114]. Alternative chemical routes include reactions between amino alcohols and $CO_2$, polyols and urea, and polyols and polyisocyanates [503]. In the last case, the polyol is normally a polyether prepared through the polymerization of propylene oxide using an initiator that contains multiple OH groups [516], as shown in Figure 26. The Covestro company claims that as many as 20 of the propylene oxide units can be replaced by $CO_2$ without significantly changing the physical characteristics of the obtained material, leading to manufacture of poly(ether carbonate) polyols [518,519]. This reaction can be catalyzed by double metal cyanide (DMC) catalysts, often Zn–Co DMC [520]. Reviews of the synthesis of polycarbonates and polyurethanes from $CO_2$ have

been presented by Peters et al. (2011) (which also includes the use of $CO_2$ for manufacture of other polymers, such as polypyrones) [158], Muthuraj and Mekonnen (2018) [460], Aresta et al. (2018) [503] and Kamphuis et al. (2019) [521].

**Figure 26.** Synthesis of polyols through the traditional reaction scheme and after replacement of propylene oxide units by $CO_2$. Reproduced with permission from Arjan W. Kleij, Michael North, Atsushi Urakawa, ChemSusChem; published by John Wiley and Sons, 2017 [516].

### 3.4.7. Oxidative Dehydrogenation (ODH) of Light Alkanes to Alkenes with $CO_2$

A significant source of olefins is shale gas, through transformation of light alkanes. Typical examples are the conversion of ethane into ethylene, propane into propylene, and ethylbenzene (EB) into styrene (the commonest dehydrogenation process that makes use of steam) [177]. Typically, the conversion of alkanes into alkenes is promoted by catalysts through direct dehydrogenation (DH or DDH). Some important commercial process include the DDH conversion of isobutane into isobutylene by Catofin using the $Cr_2O_3/Al_2O_3$ catalyst and the conversion of propane into propylene by Oleflex over the $Pt–Sn/A_{12}O_3$ catalyst [522]. However, low olefin yields are normally obtained, mainly because of catalyst coking. For this reason, oxidative dehydrogenation (ODH) constitutes an alternative route for manufacture of alkenes [522], presenting advantages associated with the less stringent thermodynamic constraints and lower rates of coke deposition on the catalysts (see Table 26).

**Table 26.** Main reactions involved in dehydrogenation mechanisms for olefin synthesis [522].

| Reaction | Stoichiometry |
| --- | --- |
| Dehydrogenation | $C_nH_{2n+2} \leftrightarrow C_nH_{2n} + H_2$ |
| $CO_2$ ODH | $C_nH_{2n+2} + CO_2 \leftrightarrow C_nH_{2n} + CO + H_2O$ |
| Mars–van Krevelen mechanism | $[O]_s + C_nH_{2n+2} \rightarrow C_nH_{2n} + H_2O + [\,]_s$ |
| | $[\,]_s + CO_2 \rightarrow CO + [O]_s$ |

$[\,]_s$ is an oxygen vacancy on the surface of the metal oxide.

ODH is usually performed with oxygen, although low selectivities toward olefins due to intense oxidation limit possible applications. Consequently, the replacement of $O_2$ by $CO_2$ as a soft oxidant has proven to be effective to increase the olefin yields [523,524]. $CO_2$ enhances the equilibrium conversion by removing hydrogen through the rWGS reaction [525], while the simultaneous overoxidation of substrates is less likely to occur due to the lower exothermicity of the reaction [523,526,527]. The Mars–van Krevelen mechanism can possibly explain the better performance of $CO_2$ as oxidant in this reaction [522] (see Table 26). Particularly, the use of bimetallic catalysts (reducible oxide-supported metal catalysts) can significantly enhance the overall process performance [527]. Gomez et al. (2019) presented an excellent review on ODH reactions performed with $CO_2$ [523]. However, before being commercially feasible, some technological problems of ODH reactions must be solved, such as the improvement of catalyst performances, optimization of reaction

conditions, proper understanding of the reaction mechanism, and optimization of the product separation step [114,523], as discussed elsewhere [523,528].

It must be emphasized that the purity of the carbon dioxide stream required by ODH reactions depends on the feed stream composition, desired products, and process flowsheet being considered [158,361]. The presence of nitrogen, for instance, affects the concentrations of reactants, affecting the reaction rates and increasing equipment size and costs [29]. Furthermore, the presence of SOx and NOx is inacceptable, as these gases promote equipment corrosion, increase the required compression values due to changes in the phase behavior, and lead to catalyst poisoning [113]. Presence of volatile organic compounds (VOCs), heavy metals (such as Hg), and hydrocarbons must also be removed to prevent the formation of undesired byproducts and catalyst poisoning [122]. Overall, most of the analyzed technologies require the use of highly pure $CO_2$ streams.

### 3.4.8. Overview of Olefin Production and Carbon Emissions

In the previous sections, several chemical routes that can be used to produce olefins (and other valuable chemicals) from $CO_2$ were presented, as summarized in Figure 27. For both ODH and OCM reactions, replacement of $CO_2$ with $O_2$ as a mild oxidant constitutes parallel research fields, which are increasing and already showing promising results. Meanwhile, MTO processes can be regarded as the most mature technologies developed so far, with some large-scale commercial plants in operation. However, the manufacture of methanol from syngas (formed by RWGS or DRM processes) still requires improvement of catalysts, process conditions, process equipment, and process layout, which are expected to evolve in the forthcoming years. Particularly, the direct formation of olefins from syngas through Fischer–Tropsch reactions has been relatively less explored, perhaps because higher selectivities toward the desired products are yet to be attained. Nevertheless, the selection of the best $CO_2$ conversion technology is highly dependent on the price and source of $H_2$, $CO_2$, and $CH_4$ [529,530].

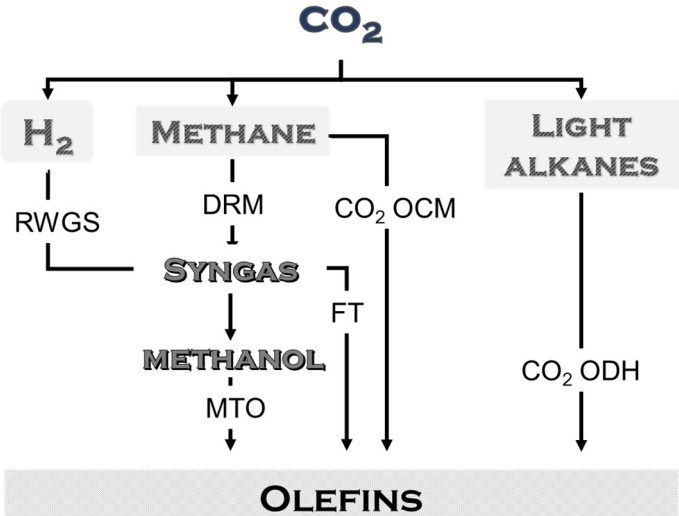

**Figure 27.** Schematic summary of the main technologies that can be used for production of olefins from carbon dioxide. Figure inspired by [38].

Regarding the inherent $CO_2$ emissions of the analyzed conversion technologies, many more studies must be performed in the near future because new advancements toward better selectivities and temperatures can modify the current technological scenarios, which has not been considered in detail. Moreover, the better comprehension of process emissions and modification of energy generation conditions can also affect the calculated $CO_2$ emissions as time goes on [529]. For instance, a study published in 2008 by Ren et al. [531] showed that steam cracking was the best technology available at that time in respect to

the reduction of the $CO_2$ emissions, but the energetic efficiency of this process and of the competitors has changed quite significantly since then. The authors considered the refence values of 0.95–1.35 kg and 0.80–1.1 kg of $CO_2$e emissions per kg of product (ethylene, propylene, and aromatics) for naphtha and ethane cracking, respectively, although these values may be as high as 1.5–2.0 kg $CO_2$e and 0.8–1.2 kg $CO_2$e per kg of olefin produced by naphtha and ethane cracking according to other studies [2,42,46,47,49,98,531,532]. Thus, although limited to China as a study case, Xiang et al. (2015) and Chen et al. (2017) proposed the MTO route as one of the most economical and environmentally friendly [514,515]. Moreover, in 2020, Keller et al. performed a life-cycle assessment (LCA), showing that production of ethylene and propylene has by far a lower global warming potential via MTO if wind electricity is used than conventional steam cracking routes [2]. Nonetheless, studies vary in results even for steam cracking (e.g., 4.2 kg $CO_2$e/kg olefins for naphtha steam cracking [515]). Therefore, more LCAs studies are needed to compare the technologies using the same scenario. However, based on the mentioned studies, methanol-to-olefin is a very promising technology. Figure 28 summarizes $CO_2$ emission values reported by different studies for distinct technologies.

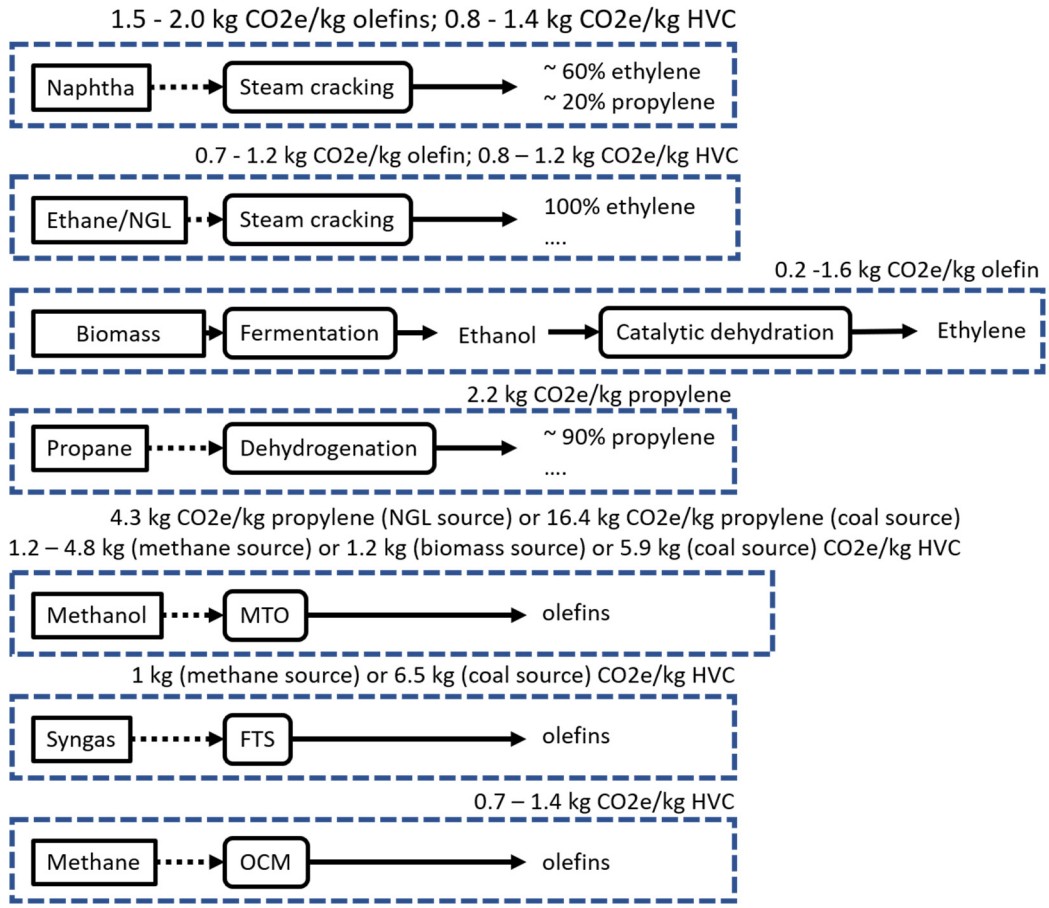

**Figure 28.** Schematic block diagram that describes the production of olefins through conventional and alternative processes. Data source: [2,38,42,46,47,49,53,74,98,514,531–533]. HVC: high-value chemicals.

### 3.5. Bibliometric Analysis of Carbon Utilization: Patents and Players

As shown throughout this text, the interest in CCU technologies is increasing. Intensive research activity is being performed, and many processes are now commercially available. Overall, for most CDU technologies, catalysts constitute the limiting technological step. Innovations are expected to improve (i) catalytic activity at certain reaction conditions, (ii) selectivities toward desired products, (iii) resistance to coke formation, and

(iv) long-term chemical and structural stability. Feed sources and hydrogen costs constitute additional limitations.

Several companies that concentrate commercial activities in different economic sectors were identified in the present study. Many startups, such as C12 Energy and Evolution Petroleum, use $CO_2$ for EOR operations. Other companies are not related directly or indirectly to the plastics business, including Air Co. (use of electrolysis to produce vodka) and C2CNT (use of electrolysis to produce carbon nanotubes). In addition, as discussed previously, the main route currently explored for manufacture of olefins and other interesting chemicals from $CO_2$ produces syngas as an intermediate for posterior catalytic Fisher–Tropsch transformation reactions. While the FT process is mature, well-known, and robust, promising routes that include $CO_2$ hydrogenation and reduction still depend on the development of more efficient catalysts. However, probably the biggest challenge that remains open is cheap $H_2$ production using renewable energy sources or renewable methane. Nevertheless, many startups are developing innovative processes to produce syngas, methanol, hydrocarbons, or other chemicals. Figure 29 and Table 27 summarize the current technological status in the field of $CO_2$ use and chemical recycling.

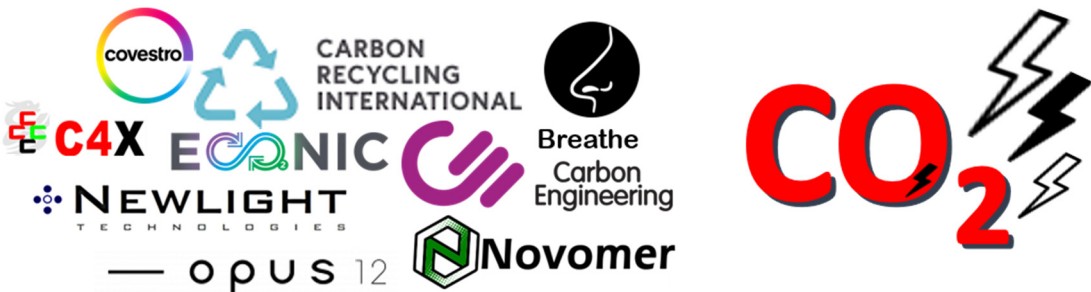

**Figure 29.** Most active companies developing technologies for $CO_2$ use and chemical recycling.

Carbon Recycling International reported the capacity to produce 4000 metric tons of methanol/year through hydrogenation, presenting the largest facility for $CO_2$ hydrogenation. The success of this plant is related to the availability of cheap geothermal electricity in Iceland, allowing the viable commercial production of $H_2$ through electrolysis of sea water. Carbon Engineering utilizes $CO_2$ for manufacture of syngas (first) and fuel (afterward) through a process named Air to Fuels™. $CO_2$ hydrogenation is also performed by Breathe, which has a stronger focus on catalysts development, and BSE Engineering (in partnership with BASF). Dimensional Energy converts $CO_2$ to syngas (or methanol) through photocatalysis, activated by the sunlight, and they developed the HI-Light™ technology to optimize the illumination of the reactor. Advanced catalysts to perform FT reactions and produce DME from syngas are being developed in collaboration by BASF and Linde [534]. Nauticol Energy reported the production of methanol through steam methane reforming (SMR), and BASF reported the production of methanol through partial oxidation of natural gas [535]. Meanwhile, dry reforming of methane has not been used on a large scale due to the current existence of technological limitations, but constitutes a promising technology for syngas synthesis, explaining why BASF is developing and commercializing catalysts for this process [536–545].

**Table 27.** Most active companies in the field of $CO_2$ utilization and respective technologies.

| Company | Country | Technology | Technology Description | Product | Scale | No. of Patents in the Field |
|---|---|---|---|---|---|---|
| AGG Biofuel | USA | Boudouard reaction; WGS | Reaction of $CO_2$ with C and steam at 1330 °C | Syngas | Pilot | 3 |
| Breathe | India | Hydrogenation | Hydrogenation of $CO_2$ from flue gas using alloys/intermetallic/bimetallic/core-shell materials based on Cu, Ni, Fe catalysts | Methanol | Bench | 0 |
| BSE Engineering | USA | Hydrogenation | Process development to use excess electrical current and off-gas $CO_2$ in parallel with catalysts development for $CO_2$ hydrogenation to methanol | Methanol | Bench | 0 |
| C4X | Canada/China | Hydrogenation of cyclic carbonates | Hydrogenation of cyclic carbonates using metal–silica catalysts | Ethylene carbonate (EC), ethylene glycol (EG), methanol | Pilot | 0 |
| Carbon Engineering | Canada | Hydrogenation | Combination of $H_2$ (split from water using renewable electricity) with captured atmospheric $CO_2$ | Fuels (gasoline, diesel, jet-A), hydrocarbons, syngas | Bench | 1 |
| Carbon Recycling International | Iceland | Hydrogenation | Production of methanol from $CO_2$, $H_2$, and renewable electricity | Methanol | Large | 4 |
| CERT (University of Toronto) | Canada | Electrochemical reduction | Development of catalysts for the reduction of $CO_2$ into CO in membrane electrode assembly (MEA) electrochemical cells, using a copper catalyst [544] | Ethylene and others | Bench | 0 |
| Covestro (Bayer spin-off) | Germany | Polymerization | Development of catalyst for producing polyether polyols and polyurethane from polyether carbonate polyol | Polyurethane | Large | 34 |
| Dimensional Energy (Cornell University startup) | USA | Photocatalytic hydrogenation | Use a photocatalyst to react $CO_2$ with $H_2$ using sunlight. | Syngas or methanol | Bench | 0 |
| Dioxide Materials | USA | Electrochemical reduction | Development of $CO_2$ electrolyzers (the Sustainion® anion exchange membranes, AEMs, based on imidazolium functionalized polystyrene) [545], and a bifunctional catalyst: combination of silver nanoparticles and ionic liquid | Syngas, formic acid, hydrocarbons | Bench | 10 |
| Econic Technologies | UK | Polymerization | Development of catalyst technologies to build polyols from $CO_2$ with epoxides and, further, polyurethane | Polyols and polyurethanes | Large | 6 |
| Hago energetics | USA | Boudouard reaction, RWGS | Conversion of $CO_2$ with C, $H_2$ (from $CH_4$), and energy [546] | Char and syngas, further processed to methanol | Pilot | 1 |
| Newlight Technologies | USA | Bio(polymerization) | Conversion of air and $CH_4/CO_2$ into biodegradable plastics by bacteria | Polyhydroxyalkanoates (PHA) | Large | 5 |

**Table 27.** *Cont.*

| Company | Country | Technology | Technology Description | Product | Scale | No. of Patents in the Field |
|---|---|---|---|---|---|---|
| Novomer | USA | Polymerization | Reaction between $CO_2$ with epoxides using a proprietary catalyst | Polyurethanes | Large | 13 |
| Sandia National Laboratories (SNL) (National Nuclear Security Administration, NNSA) | USA | Electrochemical reduction; | Electrochemical cell using an organometallic Zn; "solar reactor" (photovoltaics with an electrochemical cell) (Sandia's Sunshine to Petrol, S2P), containing reduced cobalt-doped ferrite (FeO) that absorbs oxygen, leaving behind CO and ferrite, which is recycled [547,548]. | "Liquid fuels": methanol, gasoline, jet fuel | Bench | 0 |
| Sunfire | Germany | Electrochemical reduction (power-to-methane) | Fuel cells; solid oxide electrolysis cell (SOEC) | Hydrogen, syngas, methane, methanol | Bench | 5 |
| OPUS 12 | USA | Electrochemical reduction | Membrane stack including a polymer electrolyte membrane (PEM) and catalysts | Syngas, methane, methanol, ethylene and others | Large | 2 |

According to the previous analysis, it can be concluded that most companies are using syngas as an intermediate for CDU. Carbon monoxide is produced from an electrochemical reduction of $CO_2$ or directly via the rWGS. It must be noted that the commercialization of polyurethanes synthesized from polyols constitutes a well-established market, explored by Econic Technologies, Novomer, and Covestro. Companies developing alternative processes for alkene synthesis based on dry reforming of methane (DRM) or oxidative dehydrogenation (ODH) of alkanes were not found, as these technologies are being developed mostly by more renowned petroleum and gas companies.

Patents from Plastic and Resin Manufacturers Regarding $CO_2$ Use

Once more, plastic producers have had low participation in the development of $CO_2$ utilization technologies. In the patent search performed for $CO_2$ utilization, the detected leading companies were big chemical or petrochemical companies that also produce primary plastics (see Table 28). BASF, Sabic, and ExxonMobil deposited together more than 100 patents in this field.

BASF has deposited the highest number of patents in the field: 16 related to hydrogenation, using a metal catalyst (copper or aluminum), 15 patents related to manufacture of formic acid through hydrogenation, and others related to manufacture of polyols (6), dry reforming of methane (8), methanation (3), and use of $CO_2$ as a gas stream/blowing agent (10). BASF also maintains a joint hydrogenation project with Linde to convert $CO_2$ to syngas and posteriorly to methanol [535]. Furthermore, BASF has developed the catalyst SYNSPIRE™ for use in water gas shift reactions [549,550].

Most of Sabic's patents are related to production of syngas through $CO_2$ conversion with hydrogen or methane. Thus, Sabic uses captured $CO_2$ as feedstock for production of methanol, urea, oxy-alcohols, and polycarbonates in its Jubail, Saudi Arabia, facility [551].

**Table 28.** Technology description of $CO_2$ utilization solutions developed by chemical and plastic producers.

| Company | Country | Technology | Technology Description | Product | Scale | Patents in Use |
|---|---|---|---|---|---|---|
| BASF | Germany | Polymerization, hydrogenation, FT, DRM, methanization | Formation of polyethercarbonate polyol; copolymerization of alkylene oxides and carbon dioxide using metal cyanide catalysts; hydrogenation using metal catalysts (aluminum or copper); DRM using mixed metal oxides; methanization using a ruthenium-rhenium-based or nickel/cobalt catalyst | Polyol, polyether carbonate, methanol, DME, ethylene, propylene | | 60 |
| Sabic | Saudi Arabia | Hydrogenation, FT, DRM, reduction, oxidative dehydrogenation supercritical, carrier gas, blowing agent | Hydrogenation via RWGS (and further FT synthesis) including metal catalyst (Cr, aluminia), mixed metal oxide catalysts (CuMnAl and CuZnZr); Catofin catalyst. Reduction of $CO_2$ with sulfur or metal-free catalyst. | CO, syngas, methanol, light olefins | | 46 |
| ExxonMobil | USA | Hydrogenation; Fischer–Tropsch, MTO, DRM | Using a molecular sieve to i) oxygenates conversion to olefins using also at least one metal oxide; MTO; production of syngas and, further, methanol; dry reforming of methane; RWGS | Syngas, methanol, ethylene, propylene | | 14 |
| ENI | Italy | RWGS, FT | RWGS and FT in the presence of a cobalt catalyst | Syngas; paraffins | | 5 |
| Ineos | UK | Hydrogenation; FT; DMR; blowing agent | Methanol synthesis from syngas, methane reforming to alcohols Fischer–Tropsch to olefins; blowing agent in expandable polystyrene | Methanol, ethylene, ethanol, propanol, butanol | | 4 |
| Reliance Industries | India | Fischer–Tropsch, methanol synthesis, MTO | Syngas to olefins; carboxylation reaction; dehydrogenation; gasifying agent | Olefins; nitric acid; butadiene; syngas | | 4 |
| Mitsubishi Chemical ▲ | Japan | Cyclic carbonates synthesis | Reaction between carbon dioxide and ethylene oxide | Ethylene carbonate | | 2 |
| Dow Chemical ▲ | USA | Hydrogenation; Fischer–Tropsch | Hydrogenation using a mixed metal oxide catalyst or a molecular sieve catalyst | Syngas | | 2 |
| Lotte Chemi-cals | Malaysia | Polymerization; cyclic carbonates synthesis | Copolymerization between carbon dioxide and epoxide; reaction between alkylene oxide and carbon dioxide | Polycarbonate; ethylene carbonate | | 2 |

**Table 28.** *Cont.*

| Company | Country | Technology | Technology Description | Product | Scale | Patents in Use |
|---|---|---|---|---|---|---|
| LG Chem | Korea | Polymerization | Polyalkylene carbonate resins by polymerization of carbon dioxide and epoxides | Polyalkylene carbonate | | 2 |
| PetroChina | China | Supercritical | Swelling polypropylene in supercritical carbon dioxide having dissolved vinyl monomer and an initiator | Solid phase graft modified polypropylene | | 1 |

▲ Patents of Mitsubishi Heavy Industries and Dow Global Technologies Llc were not included in the search.

Patents by ExxonMobil, Dow Chemical, Reliance Industries, and INEOS regarding CDU are also related to manufacture of syngas, methanol, and light olefins through hydrogenation and Fisher–Tropsch reactions. MTO and/or DRM technologies have also been developed by Sabic, ExxonMobil, Reliance Industries, and INEOS. It is important to emphasize that patents regarding methanol-to-olefin (MTO), oxygenate to olefin (OTO), and methanol-to-propylene (MTP) processes were not included in this frequentist analysis because these processes are normally employed after the reverse water gas shift (RWGS) or hydrogenation steps, avoiding the double counting of $CO_2$ conversion activities and inaccurate evaluation of efforts to mitigate $CO_2$ emissions, given that raw materials were not necessarily obtained through capture of $CO_2$. Otherwise, the number of patents would increase very significantly. For instance, the ExxonMobil methanol-to-gasoline (MTG) process has already been proven commercially on a scale of 700 kilotons per year [552].

Dow Chemical developed a partnership with Algenol Biofuels in 2009 to use algae for manufacture of ethanol in Freeport, Texas. In particular, Algenol's process cultivates algae inside bioreactors [553]. Dow initially planned to use ethanol produced from the pilot plant as a feedstock for manufacture of plastics (replacing natural gas), while Algenol apparently planned to use its process in coal-burning power plants. In addition, Dow Benelux (a subsidiary of Dow) is using a pilot plant to convert $CO_2$ to ethylene in the Carbon2Value project [400].

In the present search we were unable to detect ENI patents related to the production of chemicals using syngas, although the main focus of the company seems to be the manufacture of methanol using microalgae. It is interesting, however, to observe that the company is interested in producing methanol from methane and oxygen using the short contact time catalytic partial oxidation (SCT-CPO) technology, which was developed in partnership with Sabic [554,555]. In the future, the company plans to obtain methanol directly from the hydrogenation of $CO_2$ [554].

Although we were unable to detect patents related to this hydrogenation project, Mitsubishi Chemical and JGC Corp built a plant and demonstrated the feasibility of propylene synthesis from methanol or DME using a zeolite catalyst in a fixed-bed reactor (DTP® Process) [556].

Lastly, although INEOS has few patents regarding RWGS, hydrogenation, and FT processes, INEOS announced in May 2020 with INOVYN (an INEOS Business) a "power to methanol" project at Antwerp to build an industrial-scale demonstration unit that would produce 8000 metric tons per year of methanol from captured $CO_2$ combined with hydrogen generated from renewable electricity, which could be able to save at least 8000 metric tons of $CO_2$ emissions every year [557].

As also observed in the previous sections, plastic manufacturers are much less active with respect to $CO_2$ utilization than companies that actuate in other economic sectors. For instance, alarmed by the carbon bubble, several fossil-fuel companies are researching CCU technologies. As examples, Saudi Aramco and Repsol have deposited patents related to carbon dioxide conversion, especially for polymerization processes, mainly for the

production of polycarbonate and polyurethanes: (i) Saudi Aramco acquired Converge®in 2016 from Novomer, and has nine patents related to production of polyurethanes and polycarbonates [558]; (ii) Repsol has four patents related to manufacture of thermoplastic polyurethane (PU) based on polyether carbonate polyols, producing polyurethane and polyether carbonate polyols in pilot scale [559].

This scenario can be explained and justified by a number of factors: (i) the verticalization of the chemical business; (ii) the much larger market share of oil and gas companies, when compared to exclusively manufacturers of plastic resins and products; (iii) the much lower rates of $CO_2$ emissions associated with the plastics business, when compared to the oil and gas business (and energy generation). On the other hand, plastic manufacturers are being challenged everywhere in the world by the widespread feeling that most of the pollution is caused by accumulation of plastics in the environment and by stringent legal constraints and regulations. For these reasons, plastic producers are expected to become more active in the field of carbon emission mitigation, carbon capture, and carbon reutilizationFor the sake of illustrative purposes, this probably explains why we were able to detect patents regarding $CO_2$ utilization that were deposited by 13 of the 20 leading plastics manufacturers of the world.

According to the analyzed patents, the imminent readiness of the technologies discussed previously for $CO_2$ conversion and FT transformations seems clear. In general, most of these technologies are not yet employed on large commercial scales; however, by the time hydrogen and methane become more readily available, $CO_2$ use and chemical recycling are expected to increase beyond the manufacture of polycarbonates and polyurethanes. The use of $CO_2$, normally regarded as a byproduct or waste, within the factory reduces GHG emissions and the need for utilization of fossil fuels. However, for this scenario to become real, hydrogen and methane must be obtained from green sources. The fact that these technologies are almost commercially available should encourage plastic companies to invest in these technologies as well.

## 4. Conclusions, Challenges, and Future Trends

Plastics are prominent, versatile, and sustainable materials, whose production is expected to increase 3% per year [4] (5.5% for the packaging industry even with Covid-19 [560]). However, adjustments are needed not only to solve the final disposal problem but also to reduce the GHG emissions of plastic manufacturers. In the circular economy scenario, the recycling rate must be increased and CCU technologies must be employed simultaneously, to prevent the release of carbon in the atmosphere, aqueous environments, and land.

When compared to other chemical industries, plastic production is among the most energy-demanding. Most of the emitted $CO_2$ can be associated with energy consumption and heat generation, although direct industrial emissions occur during oil and gas cracking, hydrogen production, and feedstock manufacture. According to this scenario, it is surprising to observe that GHG emissions from plastic producers have been largely neglected by published scientific works. As a matter of fact, most academic publications regarding the carbon footprint of plastic industries were published in the last 5 years [2,7,35–37,42,43,48,58,93,159,481,517,529,531,561], while, in other economic segments (oil and gas, steel, cement, and ammonia, among others), GHG pollution data have been reported more frequently for several years. This shows that, although $CO_2$ emissions have not been discussed as intensively in the field as plastic recycling, the awareness of $CO_2$ emissions from plastic production is increasing, pressing plastic companies to fully incorporate circular economy strategies and reduce their environmental footprint.

In this review, challenges and opportunities for carbon capture and utilization were presented and discussed, focusing more specifically on the interests of the plastic industry. Although some of the analyzed technologies are still at the laboratory-scale stage of development, this should not discourage investments on CCU, as advancements are being introduced at accelerated rates and increasing cost-effectiveness. On the other hand, carbon

capture technologies have been investigated for a long time, and many of them are now commercially available. If carbon is to be captured within the post-combustion framework, the use of amine-based absorption constitutes the most widely known, mature, and used technology. Chemical absorption is also adequate for other direct process emissions where $CO_2$ is available at low partial pressures, which is the common scenario in most plastic industries. However, due to the high energy demand of chemical absorption, modern plants are replacing the classic MEA and DEA with new solvents. Ionic liquids, piperazine, and piperazine derivatives are some possible options. Consequently, the introduction of new solvents will probably maintain the commercial attractivity of chemical absorption processes for carbon capture in the near future.

In addition, other carbon capture and purification processes can be regarded as promising technologies. An example is adsorption, as it can be easily retrofitted with relatively high capacity and $CO_2$ selectivity [202]. Other viable technologies, although not yet applied commercially at a large scale, are membrane separations, albeit their current applications are only feasible when the $CO_2$ composition is greater than 0.2 [163,562,563]; and electrochemical transformations, which are receiving high investments and advancing very fast [452], as these processes can simultaneously react $CO_2$ and synthesize high-valued products, such as formic acid, CO, methanol, and methane [478]. Other alternatives for $CO_2$ capture, such as mineralization and bioconversion, do not seem attractive for plastic manufacturers, although the conversion of biomass into high-valued compounds through biorefining certainly constitutes an important trend for the future of plastic manufacturers [564–566].

It is important to recognize that carbon capture comes with a cost. For this reason, it is expected that new government incentives and more stringent laws and regulations that favor the sustainable chemical production and decarbonization of the economy will encourage the use and development of greener technologies in the near future. Simultaneously, it is also expected that the continuous development of technical studies and optimization of analyzed technologies will eventually lead to implementation of economically viable green processes [567]. Particularly, commercial projects, such as the Carbon Recycling International in Iceland, prove that sustainable production is possible when green energy is available [2,561]. Furthermore, the idea that captured $CO_2$ should be stored or used for natural gas/oil extraction is popular, but as $CO_2$ is an attractive building block, the use of $CO_2$ as a feedstock to produce valuable products must be emphasized, helping simultaneously to offset carbon capture costs [362].

The CDU main products are methanol (and other products derived from it), methane, olefins, polycarbonates, and polyurethanes. The fixation period of $CO_2$ stored in these chemicals depends on the pursued application and molecular stability of product. From these main products, plastics are certainly the best materials to store carbon for longer periods. In contrast, the use of CDU processes to manufacture oxygenates (ethanol or DME) to be used as fuels ($CO_2$-to-fuels) releases the stored $CO_2$ very rapidly to the atmosphere, although this is the biggest market for $CO_2$ utilization. If ethanol is used for manufacture of olefins instead, the fixation period of $CO_2$ can certainly become much longer. The idea of long $CO_2$ storage periods is similar to those used by companies that produce polycarbonates and polyurethanes. In summary, the use of $CO_2$ as a raw material can constitute an enormous source of feedstock and sustainable opportunities for the plastics industry. The most interesting chemical route for conversion of $CO_2$ is currently hydrogenation, despite its relatively high costs and the necessity to develop more efficient catalysts. However, if cheap and renewable sources of hydrogen become available, immediate commercial implementation of hydrogenation processes will be feasible. Meanwhile, the use of methane from stranded gas or from biogas constitutes an excellent alternative for $CO_2$ conversion (via dry reforming of methane) [411,487].

Nevertheless, Figure 30 shows that many plastics companies are not yet active in the CDU field, although BASF and Sabic are the most active in $CO_2$ utilization. BASF runs the Carbon Management Program that considers several strategies to reduce the GHG

emissions: hydrogenation, methane to olefins, electrification, renewable energy, and plastic recycling, among others [568,569]. Sabic is capturing carbon from commercial ethylene oxide industrial plants and investing in green energy, electrification, and plastic recycling [570]. Meanwhile, Reliance Industries claims that green energy and $CO_2$ utilization are part of the company plans [571]. ExxonMobil, Sumitomo Chemical, and INEOS are investing in plastic recycling [572–576]. ENI is developing microalgae biofixation processes to fix $CO_2$ and announced in 2020 the creation of a new business group called "Natural Resources" for the development of carbon capture technologies [577,578]. Dow claims that the company will be investing in renewable energy, plastic recycling, and CCU technologies [579,580]. Perhaps the larger number of patents and announcements from these companies is associated with their broad range of petrochemical products and urgency to adapt to global demands.

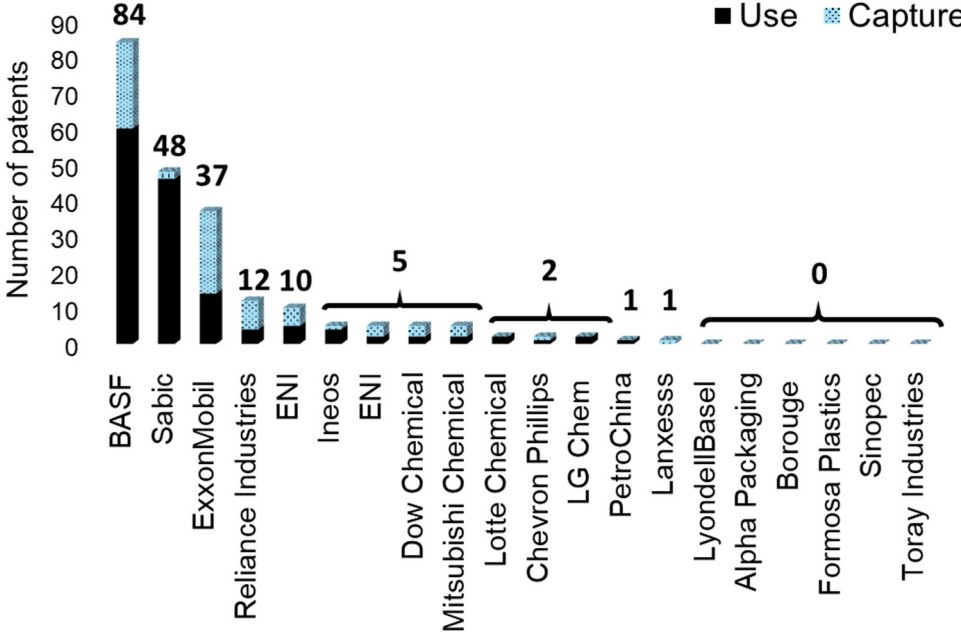

**Figure 30.** $CO_2$ capture and use patents for the investigated plastic producers (2002–2020).

As a matter of fact, most $CO_2$ capture and use technologies are not related to plastic facilities. Surely, plastic recycling is a very important initiative, but not sufficient to reduce $CO_2$ emissions. For this reason, plastic recycling and CCU technologies should be carried out simultaneously. This scenario can be explained and justified by a number of factors, including the verticalization of the chemical business, the much larger market share of oil and gas companies, and the much lower rates of $CO_2$ emissions associated with the plastics business, when compared to the oil and gas business (and energy generation). Nevertheless, plastic producers are expected to become more active in the field of carbon emission mitigation, carbon capture, and carbon reutilization to face the current challenges of the business.

Up to now, one of the best initiatives to mitigate $CO_2$ emissions in the plastic chain is the "Cracker of the Future" electrification project organized by BASF, Borealis, BP, LyondellBasell, SABIC, and Total. Following this project, investments in energy efficient practices and generation of renewable energy are also occurring [25]. However, electrifying the cracker means redesigning the whole furnace (alloy composition, electric connectors, and transformers), meaning that existing crackers will not be electrified instantaneously. Despite that, this is a significant technological drive for the installation of more sustainable crackers in the forthcoming years and a significant reduction in the carbon footprint of the plastic business.

The main idea behind this review was to present technologies that can be used to capture carbon in plastic industries and utilize the captured $CO_2$ through appropriate process integration [28,187]. The large number of papers and patents associated with carbon dioxide recycling illustrates how rapid and important research and development is in this field. Overall, further developments of carbon dioxide recycling will require contributions from various scientific fields, including industrial partners and policymakers [50]. Fortunately, CCU and CDU technologies are being developed quickly, and they can be rapidly implemented in this world faded with a climate-induced collapse, despite the many technological challenges discussed throughout the text. Lastly, although there are no doubts that CCU and CDU technologies offer many opportunities for the plastic industry, plastic manufacturers must get more deeply involved with CCU and CDU technologies in the forthcoming years.

**Author Contributions:** Bibliometry, L.P.d.M.C., D.M.V.d.M. and A.C.C.d.O.; document research, L.P.d.M.C., D.M.V.d.M. and A.C.C.d.O.; discussion, L.P.d.M.C., D.M.V.d.M., L.F., M.S.S.P., S.J.W., M.H.S.A. and J.C.P.; writing, L.P.d.M.C., D.M.V.d.M. and J.C.P.; revision, L.P.d.M.C., D.M.V.d.M., A.C.C.d.O., L.F., M.S.S.P., S.J.W., M.H.S.A. and J.C.P.; funding, L.F.; project coordination, L.F. and J.C.P.; industrial data, L.F., M.S.S.P. and S.J.W.; technical coordination, M.S.S.P.; information management, M.H.S.A.; I.G.B.: industrial data, discussion, revision; supervision, J.C.P. All authors have read and agreed to the published version of the manuscript.

**Funding:** This research was partially supported by Braskem S.A. [Contract 4600016591/2018]. The authors thank Conselho Nacional de Desenvolvimento Científico e Tecnológico (CNPq) and Coordenação de Aperfeiçoamento de Pessoal de Nível Superior (CAPES) for providing scholarships to LPC, DVM, ACCO and JCP.

**Institutional Review Board Statement:** Not applicable.

**Informed Consent Statement:** Not applicable.

**Data Availability Statement:** All collected data are presented in the manuscript.

**Conflicts of Interest:** The authors declare no conflict of interest.

## Appendix A

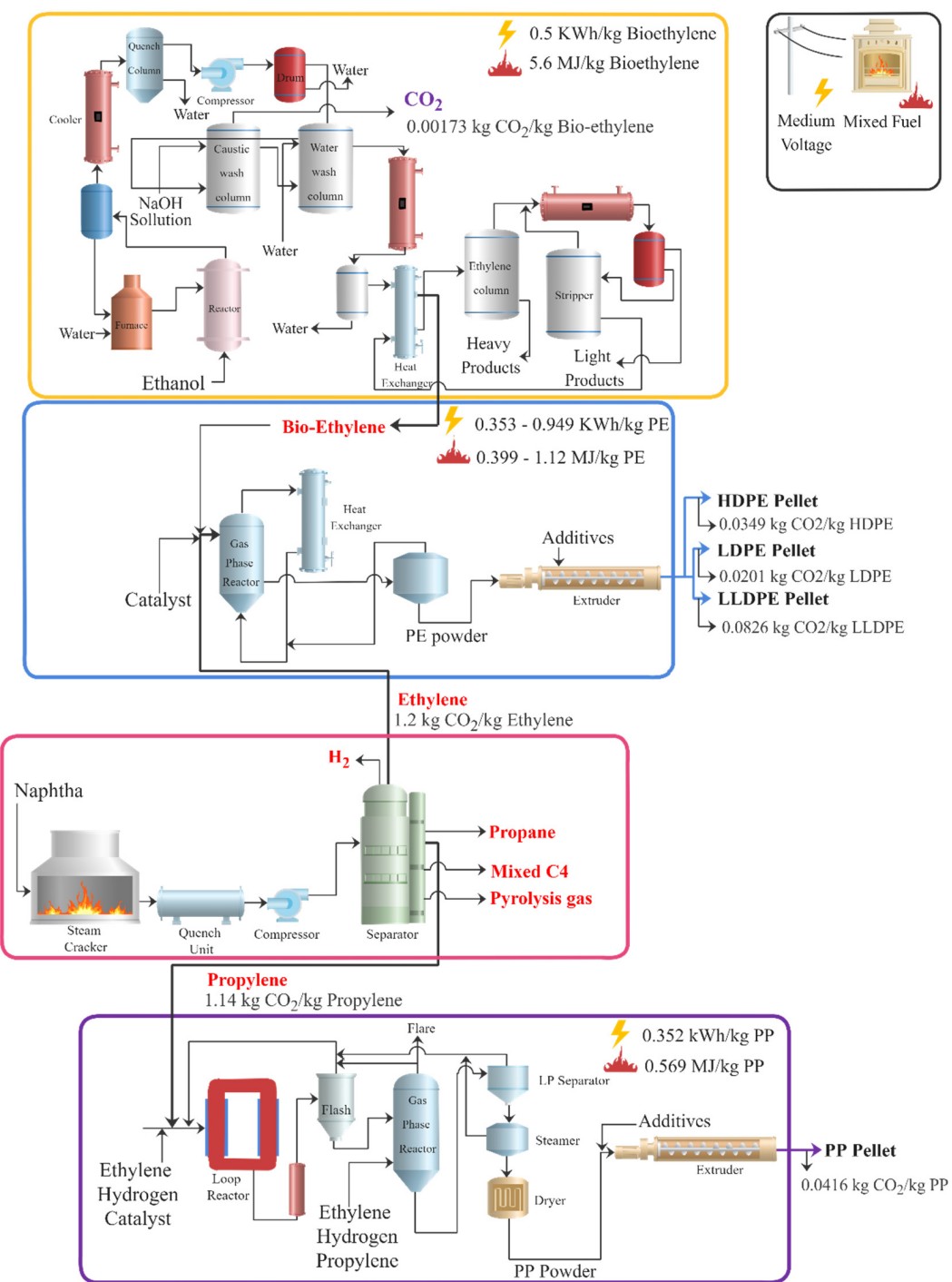

**Figure A1.** Detailed energy consumption and $CO_2$ emissions for polyethylene and polypropylene production chains (data from the Ecoivent 3.6 dataset [56]).

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
