# Peer review of "Capture and Reuse of Carbon Dioxide (CO2) for a Plastics Circular Economy: A Review"

_processes, doi:10.3390/pr9050759_

Round 1

Reviewer 1 Report

Overall, this is an impressive study in terms of data quantity, number of tables, figures and references used. In my opinion, it's looks more like an introduction to a PhD thesis, than a review paper. Nevertheless, it would be a very good source for the future readers interested to find the main examples directed towards CO2  capture and reuse. Indeed, plastics are here to stay (due to their many advantages), but we must work together for a better management of plastic waste, and pollution.

All in all, I found this extensive paper well-written, with very few inconsistencies (some missing links between text and tables, and a few places where the font was smaller - see the attached .pdf).

I didn't see any financial statements nor conflict of interests forms (although several authors are currently employed by Braskem - one of the largest petrochemical players in South America).

Author Response

Dear Reviewer,

We thank your contribution to our work! Indeed, many topics are explained in a very introductory manner. But, as you have pointed out, and because GHG emissions are not discussed widely within the plastic industry (and academically as well - at least between polymer engineers!), we decided that we would like to discuss the basics of each CCU technology so that one, never in contact with the technologies, establishes a baseline understanding.

Overall the review aims to i) have an overview of each technology, ii) address their development according to the number of studies and patents; iii) address which are already being chosen for implementation at large scales; iv) how they are being considered for primary petrochemical producers up to now.

Regarding the inconsistencies, we have accepted them all! Thanks! Attached you will find the new version which we hope to have corrected all of the mistakes found up to now. We have also included a conflict of interest and a funding statement.

Once more, thank you for your contribution! We hope the concerns were solved.

Reviewer 2 Report

This article reviews literature pertinent to CO2 capture and utilization in plastic production industry. Given the vast available and relevant information, the authors completed a very good job in developing a methodology to present the most updated information in a well-organized manner. In particular, given the fast-growing nature of CCU studies and efferts in the world, this methodology seems very appropriate. Moreover, the authors’ endeavor in finding and categorizing the information on the companies and patents pertinent to CCU is admirable and unique. My only problem with this manuscript is that, although the focus of the study is supposed to be on CCU in plastic production industry, more than 80 percent of the text covers the topic in general manner. Perhaps it would be nice if the authors somehow remove or soften this claim in the title, abstract and conclusions. It would be also very nice to reduce the size of the manuscript by being more concise, as I observed some some information was repeated throughout the paper.  

Unfortunately, the line number was not available, but I try to mention a couple of other points here:

In part 1.1 please correct Error! Reference source not found.

Could you represent Figure 3 in a neater and more simple way?

The details in Figure 10 are too small to be read and understood.

Table 7 and 10 present a lot of similar information. Could you merge them into one?

Page 26: CO2 is not always adsorbed physically, for instance it is chemically adsorbed on supported amine sorbents via carbamate and bicarbonate formation.

In the conclusions part membrane was listed as the second viable method after absorption, omitting adsorption as the method that is more widely practiced industrially.

Author Response

Dear Reviewer,

We thank your contribution to our work! Indeed, many topics are explained in a very introductory manner. However, as GHG emissions are not discussed widely within the plastic industry (and academically as well - at least between polymer engineers!), we decided that we would like to discuss the basics of each CCU technology so that one, who never study these topics before, can establish a baseline understanding.

Overall, the review aims to i) have an overview of each technology, ii) address their development according to the number of studies and patents; iii) address which are already being chosen for implementation at large scales; iv) how they are being considered for primary petrochemical producers up to now.

Regarding the inconsistencies, we have accepted them all! Specifically, we have adapted Figure 3 and added the previous one to the Appendix; and we removed Table 10. In addition, about the error on page 26, you are correct! We have also added adsorption as a viable option in the conclusion as it truly is! For instance, throughout our patent search, we have found many patents related to adsorption – published the last years, I am remembering many using MOFs. Attached you will find the new version which we hope to have corrected all of the mistakes found up to now.

Once more, thank you for your contribution! 
